# *Drosophila* uses a tripod gait across all walking speeds, and the geometry of the tripod is important for speed control

Chanwoo Chun[1], Tirthabir Biswas[2,3], Vikas Bhandawat[4]*

[1]Department of Biology, Duke University, Durham, United States; [2]Department of Physics, Loyola University, New Orleans, United States; [3]Janelia Research Campus, Howard Medical Institute, Ashburn, United States; [4]School of Biomedical Engineering, Sciences and Health Systems, Drexel University, Duke Institute for Brain Sciences, Duke University, Durham, United States

**Abstract** Changes in walking speed are characterized by changes in both the animal's gait and the mechanics of its interaction with the ground. Here we study these changes in walking *Drosophila*. We measured the fly's center of mass movement with high spatial resolution and the position of its footprints. Flies predominantly employ a modified tripod gait that only changes marginally with speed. The mechanics of a tripod gait can be approximated with a simple model – angular and radial spring-loaded inverted pendulum (ARSLIP) – which is characterized by two springs of an effective leg that become stiffer as the speed increases. Surprisingly, the change in the stiffness of the spring is mediated by the change in tripod shape rather than a change in stiffness of individual legs. The effect of tripod shape on mechanics can also explain the large variation in kinematics among insects, and ARSLIP can model these variations.

*For correspondence:
vb468@drexel.edu

## Introduction

Behavior, including locomotion, results from interactions between the nervous system, the body, and the environment (*Chiel and Beer, 1997*; *Full and Koditschek, 1999*). Despite a history of research in both neurobiology (*Büschges et al., 2008*; *Cruse, 1990*; *Delcomyn, 1985*; *Dürr et al., 2004*; *Graham, 1985*) and biomechanics (*Full and Koditschek, 1999*; *Full and Tu, 1990*), a complete integration of neural and mechanical systems for legged locomotion remains elusive. Recent developments in both methods for assessing neural activity (*Maimon et al., 2010*; *Seelig et al., 2011*; *Wilson et al., 2004*) and the vast and ever-improving genetic toolkit *Venken et al., 2011* have made *Drosophila* a vital model system for the study of neural control of behavior. In contrast, the mechanics of legged locomotion in flies remains understudied. In this study, we will focus on changes in speed during walking: we will first describe interleg coordination (used interchangeably with gait in this article), a necessary first step toward understanding mechanics, and then the mechanics of body–environment interaction that accompany changes in speed.

In insects, changes in interleg coordination with change in speed are strikingly different from mammals: mammals undergo transition from walking to other coordination patterns such as run, trot, or gallop at precise speeds. Moreover, in mammals, gait transitions measured in terms of speeds relative to their size defined as Froude number (*Fr*) occur at specific *Fr* (*Alexander, 1989*). They walk below *Fr* of 0.3 while choosing other gaits at higher *Fr*. In contrast, insects employ a tripod gait at a wide range of *Fr* from 0.001 in flies (*Biswas et al., 2018*), *Fr* of 0.25 in ants (*Reinhardt et al., 2009*), and *Fr* > 1 in cockroaches. Insects do change their gaits (*Wendler, 1966*); when insects change gait, the gait selection in insects appears to be probabilistic, that is, different

gaits can be employed at the same speed. Regardless, tripod is the most common gait in insects and why a tripod coordination can support a large range of speed is not well understood.

Changes in speed are also accompanied by changes in mechanics and are particularly well understood in mammalian locomotion. In particular, the mechanics of the center of mass (CoM) during locomotion are relatively simple, and models to explain CoM mechanics have provided many insights (*Dickinson et al., 2000*; *Full and Koditschek, 1999*). During mammalian walking, the CoM is at its highest position at mid-stance, and the horizontal speed of the CoM is lowest at mid-stance (*Figure 1A*). Running in humans or galloping in quadrupeds displays different kinematics from walking that is characterized by a minimum in CoM height (*Figure 1A*). Both the walking and running CoM kinematics can be explained by a simple mechanical model called the spring-loaded inverted pendulum (SLIP). In the SLIP model, the mass of the animal is concentrated into a point mass, which is supported by a single, massless effective leg (*Figure 1A*). During the first half of stance, the spring is compressed as the body moves through the stance phase, converting kinetic energy into elastic energy stored by leg muscles and tendons. During the second half of stance, the stored elastic energy is converted back into kinetic energy. Thus, the kinetic energy, and therefore the speed,

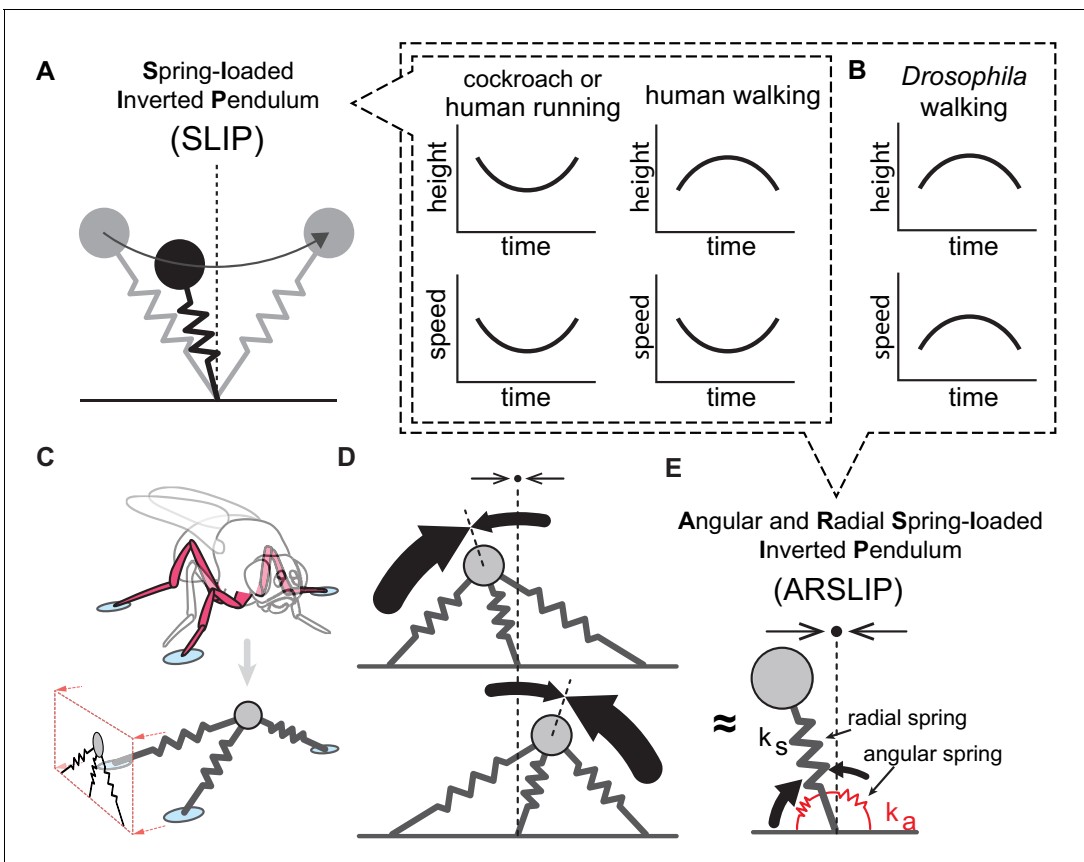

**Figure 1.** A new mechanical model for a tripod gait. (**A**) Schematic showing a simple model for the center of mass (CoM) movement during locomotion. In this model (spring-loaded inverted pendulum [SLIP]), the mass of the animal is concentrated into a point that is supported by a single massless spring. The arrow represents the direction of locomotion. This model can capture the basic features of the CoM movement during walking (in humans) and running (in both humans and cockroaches). (**B**) SLIP cannot describe the fly's CoM movement, which has a mid-stance maxima in speed. (**C**) A fly walking on three legs can be described by a springy tripod. The sagittal plane mechanics is governed by the sagittal plane projection of the springy tripod (see rectangle). (**D**) A springy tripod will produce angular restorative forces. Any movement away from the mid-stance position will produce restorative forces (represented by thin arrows). The thick arrows represent forces from front and back legs; thickness of the arrow indicates the magnitude of the force. Top: before mid-stance, the front leg is stretched and the back leg is compressed, leading to larger forces from the back leg. Bottom: after mid-stance, the front leg is compressed and exerts larger tangential forces. Net result is restorative forces. (**E**) The sagittal projection of a springy tripod can be modeled as the angular and radial spring-loaded inverted pendulum (ARSLIP) model. The angular springs expand as the CoM moves away from the mid-stance position and thereby generate restorative forces. The restorative forces can produce the mid-stance maximum in speed observed in flies in addition to the CoM movement pattern in human walking and running.

reaches its lowest value at mid-stance, as does the height in most cases. These mid-stance minimum in speed and height are also observed during running in many different mammals (*Blickhan, 1989*; *Blickhan and Full, 1993*; *Cavagna et al., 1977*; *McMahon, 1984*; *McMahon and Cheng, 1990*), making SLIP an effective model for running. More recently, it has been appreciated that SLIP can also serve as a model for walking by producing a speed minimum and height maximum at mid-stance (*Figure 1A*) when the spring is stiff (*Geyer et al., 2006*). That SLIP can serve as a model for both walking and running has proven useful as a unifying model for mammalian locomotion.

SLIP can also serve as a model for running in cockroaches. An elegant series of studies on running cockroaches has shown a striking similarity to mammalian running; in both cases, the CoM reaches a minimum in speed and height at mid-stance (*Full and Tu, 1990*, *Full and Tu, 1991*) and can be modeled by SLIP. The three legs of a tripod can be replaced by a single spring-loaded effective leg. However, by its very nature, SLIP cannot generate the CoM kinematics of many insects including *Drosophila* (*Figure 1B*) because a fly's horizontal speed during walking is at its maximum at mid-stance (*Graham, 1972*; *Mendes et al., 2013*). Therefore, a mechanical framework consistent with both the CoM kinematics in flies and cockroaches is necessary.

A qualitative consideration of the mechanics of an animal walking on three legs shows that SLIP might be an oversimplified model: an animal walking with a tripod gait can be approximated as a point mass supported by three massless springs or a springy tripod (*Figure 1C*). The sagittal plane projection of the springy tripod shown in the red box in *Figure 1C* is the mechanical system that governs the CoM movement in the sagittal plane. A springy tripod cannot be approximated by SLIP because the springy tripod is stable while SLIP is unstable. An animal supported by a single SLIP-like leg will fall. As the CoM moves away from the vertical (say toward the front of the fly), the front leg compresses and tends to push the fly backward (*Figure 1D*). Similarly, if the CoM moves back, the hind leg will push it forward. These restorative forces cannot be modeled by SLIP but can be modeled by a simple extension to SLIP through the addition of an angular spring to model restorative forces. In other words, the three legs of a tripod act like a single leg whose behavior is described by a new biomechanical model – angular and radial spring-loaded pendulum (ARSLIP; *Figure 1E*). This model would enable the modeling of both the cockroach-like and fly-like kinematic patterns.

The mechanics of a springy tripod is not only affected by the stiffness of individual legs but also by its geometry (or where the legs are positioned on the ground). Changes in geometry can be a mechanism to accommodate large variation in speed supported by the tripod gait and have the potential to explain why tripod gaits can support a large range of speeds. The effect of geometry on mechanics can also be modeled by the ARSLIP model through the differential effects of the geometry on the two spring constants – the radial and angular spring constants (*Figure 1E*) – which describe the ARSLIP model.

In this study, we created an automated method for measuring the movement of a fly's CoM in all three dimensions while also tracking the position of the fly's stance legs. Using this method, we analyzed a fly's gait over >500 steps during which the fly is always walking straight. Flies employ a modified tripod (M-tripod) gait throughout their entire speed range with only a small dependence on speed. The proposed ARSLIP model can explain how tripod geometry affects the nature of forces that act on the fly, and ultimately defines its dynamics and can provide an elegant explanation for why insects do not change their gait over a wide speed range.

## Results

### An automated method for obtaining a fly's walking kinematics with high spatial resolution

We designed an automated data acquisition system that generates a large positional dataset with high spatial resolution to investigate the fly's gait and CoM kinematics. Similar to an approach employed previously (*Nye and Ritzmann, 1992*; *Wosnitza et al., 2013*), we recorded the side and the bottom (reflected off a mirror) view of a fly walking in a clear, closed cuboid chamber (*Figure 2A*). We extracted all the steps during which a fly walked straight for more than one step. The fly's CoM was extracted using the Kanade–Lucas–Tomasi (KLT) (*Tomasi and Kanade, 1991*) algorithm and produced low-noise estimates of the CoM position; the vertical resolution being 20 μm (see 'Materials and methods', 'Tracking CoM and foothold positions', *Video 1*), which makes the

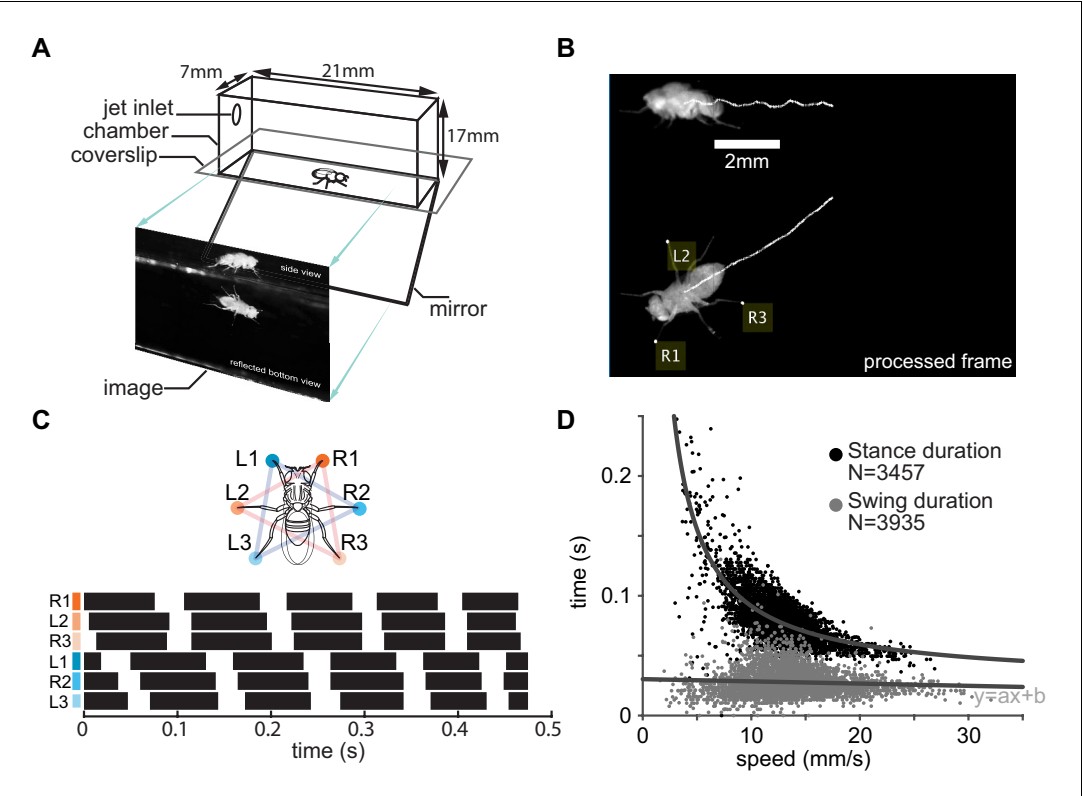

**Figure 2.** Experimental setup. (**A**) Schematic of the arena. (**B**) A frame from a typical processed video of a fly walking in the chamber. The white traces track a point on the thorax and are a proxy for the center of mass. Yellow labels denote feet that are in stance, or the footholds. (**C**) Leg numbering and color coding (top), and gait maps (bottom) showing footfall patterns of individual legs in the R1–L2–R3 (orange) and L1–R2–L3 tripods (blue). Each row corresponds to a single leg. Black bars represent stance. (**D**) Stance and swing durations as functions of speed. The top dark gray line is the best fit of the reciprocal function to stance durations. The bottom lighter gray line is a best fit of linear regression to swing durations and shows a small decrease (a: −0.00018, p<0.00001, b = 0.03).

The online version of this article includes the following figure supplement(s) for figure 2:

**Figure supplement 1.** Image processing used to obtain the 3D coordinates of the center of mass (CoM) and the time series of footholds in the body coordinate system (referred to from 'Materials and methods').

**Figure supplement 2.** Size of experimental tracking error compared to height change and speed change values (referred to from 'Materials and methods').

rhythmic up-and-down movement of the CoM apparent (*Figure 2B*). The positions of the leg tips during stance were extracted using a custom algorithm (see 'Materials and methods', 'Tracking CoM and foothold positions', *Figure 2—figure supplement 1*). The legs were labeled according to an established convention (*Figure 2C*), and the gait map (*Figure 2C*) was put together such that the legs that constitute a tripod – right prothoracic (R1), left mesothoracic (L2), and right metathoracic (R3) – are plotted on consecutive rows (orange); and those of the other tripod (L1–R2–L3) are plotted in another set of consecutive rows (marked in blue), to allow a direct assessment of the presence or absence of the tripod gait.

As a means of corroborating previous findings, we plotted stance and swing duration as a function of speed (*Figure 2D*). Consistent with previous studies (*Graham, 1972*; *Mendes et al., 2013*; *Pearson, 1976*; *Strauss and Heisenberg, 1990*; *Wilson, 1966*), the stance duration is inversely proportional to speed. The swing duration also changes with speed but to a smaller extent than the changes in stance duration.

## Flies employ interleg coordination close to tripod across speeds

We used two methods to characterize the speed-dependent change in coordination between legs: first, to facilitate comparison with previous work, interleg coordination was defined based on delays

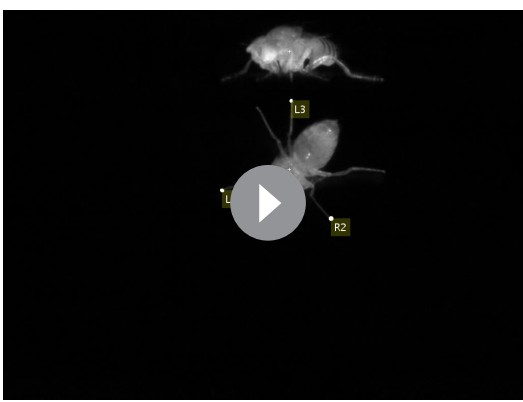

**Video 1.** Video of the fly showing the top and side views. The center of mass is marked on both the top and side views. The rhythmic up and down movement of the center of mass can be seen on the top view. The stance legs are also marked.
https://elifesciences.org/articles/65878#video1

between the times at which the legs start either a swing or a stance phase. To visualize a fly's gait, the times that a fly's legs start the stance phase in relation to the time that the right front leg (R1) entered the stance phase were plotted. Legs that form the first set (R1–L2–R3) of tripod legs enter stance phase with a short interleg delay (*Figure 3A*). The legs that form the other tripod (L1–R2–L3) enter the stance phase with a short interleg delay with each other but out of phase with the first set. The coordination pattern did not change noticeably as a function of speed (*Figure 3A*). This raw gait map (*Figure 3A*) suggests that the flies predominantly employ a tripod gait across all speeds. This trend (in *Figure 3A*) was quantified by calculating the delays relative to the cycle period (the time it takes a leg to complete both a swing and a stance, *Figure 3B*) or normalized delay. The normalized delays between the legs of the same tripod were small throughout the entire speed range; the within tripod delays became even

smaller with speed. The prothoracic leg led the other legs in its tripod with a small but significant negative delay consistent with previous observations in cockroaches (*Bender et al., 2011*; *Delcomyn, 1971*). On the other hand, the normalized delays across legs in the opposing tripods were 0.5 (*Figure 3B*). These analyses suggest that the gait – as defined by phase differences between legs – employed by flies during forward walking across the entire range of speeds is close to a tripod.

The normalized delays between different legs are consistent with that of a single gait that is close to a tripod but in which the front leg of the tripod is ahead of the middle leg, which in turn is ahead of the rear leg; we will refer to this gait as M-tripod (*Figure 3C*). The delays between legs within a tripod do decrease slightly with speed (*Figure 3C*). The small dependence means that there is no qualitative change in gait. This small dependence on speed is consistent with the continuum of coordination patterns observed in a recent study (*DeAngelis et al., 2019*).

A second method to quantify leg coordination is to use instantaneous phase lags between legs (*Figure 3D*) averaged over a gait cycle (*Couzin-Fuchs et al., 2015*; *Revzen and Guckenheimer, 2008*). Although time delays are easier to visualize and phase lag more abstract, the latter provides a more accurate measure of coordination because it takes the entire step into account instead of only the beginning of stance (see 'Materials and methods' 'Gait analysis based on leg phases'). As in the case of stance start times, the distribution of phase lags between the reference leg (R1) and the other legs show small phase differences between the legs within a tripod and large phase differences between the legs in the opposing tripod (*Figure 3E*). The phase plots also reveal that the front leg of the tripod leads the middle and back legs. As the speed increases, the phase difference between the tripod leg decreases, and the spread of the phase difference also becomes smaller (*Figure 3E*). The analysis using instantaneous phases is consistent with a single gait – M-tripod – across the entire range of speed; the exact speed dependence of M-tripod gait using phase difference (*Figure 3F*) is slightly different from the speed dependence calculated from stance start times (*Figure 3C*).

A small percentage (about 4%) of steps at very low speeds did not conform to any gait, and a few steps had a tetrapod coordination pattern, but an overwhelming majority of the steps have a tripodal coordination. The rest of the study will focus on steps that have a tripodal coordination.

## Kinematic changes associated with changes in speed

Given that flies can walk over their entire speed range using a M-tripod gait implies that a change in gait is not essential for a change in speed. To better understand the mechanism underlying change in speed, we focused on the tripodal steps and asked how the movement of the fly's CoM over the tripod gait cycle changed with speed. Because the tripod legs are not perfectly in sync, we defined

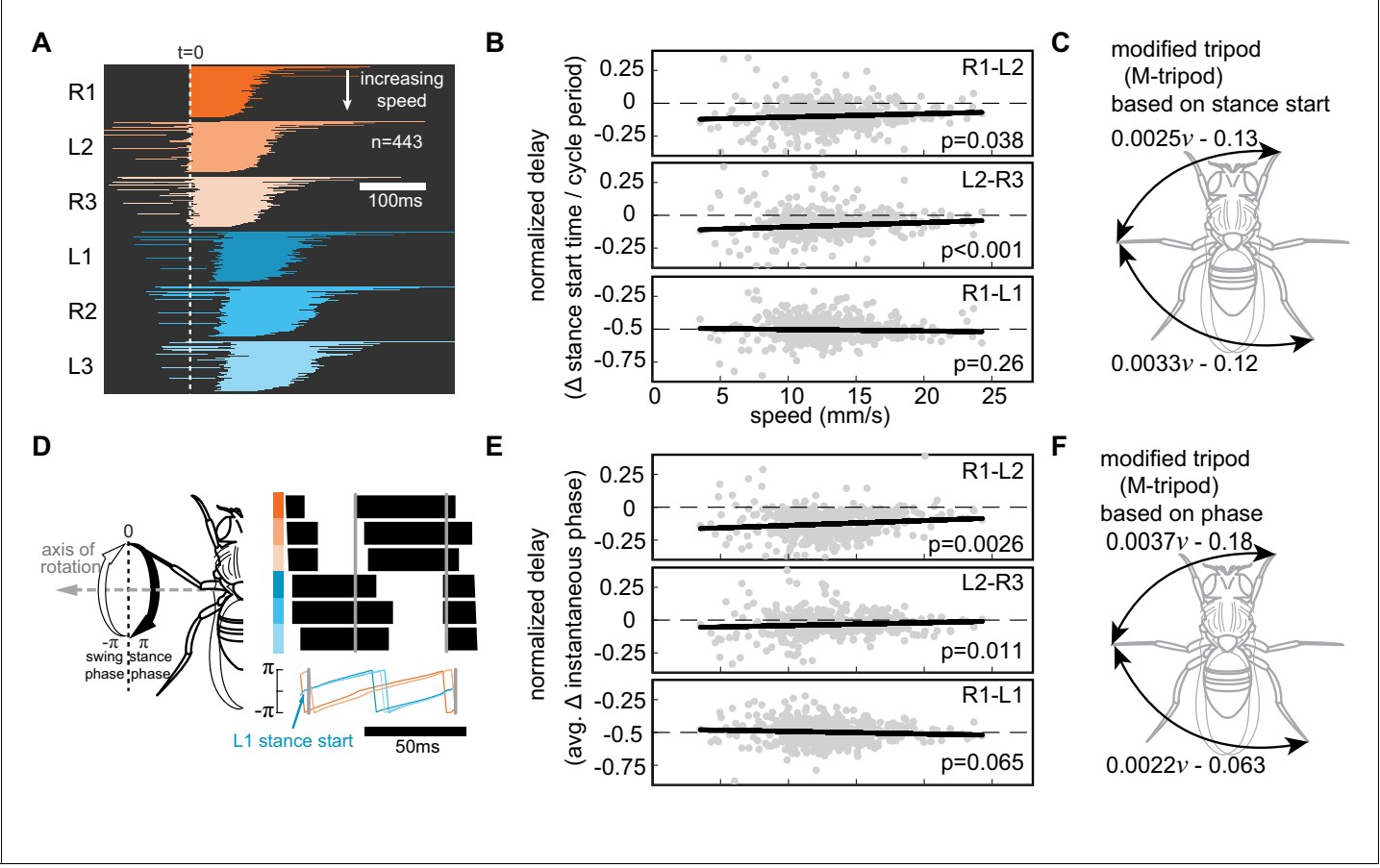

**Figure 3.** Interleg coordination pattern is consistent with a modified tripod (M-tripod) gait. (**A**) Stance for all steps relative to R1 sorted by speed. (**B**) Normalized time delays of stance start times between legs within a tripod (R1 and L2, L2 and R3) and legs in the opposing tripod (R1 and L1). The time delays were normalized by cycle duration. R1–L2 and L2–R3 delays are small at low speeds and become even smaller as the walking speed increases (Wilcoxon sign-rank test). R1–L1 delays are unchanged. (**C**) The phase difference between legs is consistent with a single gait, which is a modified version of a tripod (M-tripod) in which the front leg of the tripod leads the middle leg that in turn leads the back leg. The delay between the legs has a small dependence on speed (v). (**D**) Definition of leg phase angles. Stances start and end at 0 and π, respectively; swings start and end at –π and 0, respectively. (**E**) The leg phases relative to R1 show that interleg coordinations at different speeds all consistent with M-tripod. The delays between tripod legs do become smaller with speed (Wilcoxon sign-rank test) while the delays between R1 and L1 leg remain unchanged. (**F**) The M-tripod based on phase lag (v=speed).

the tripod start as the halfway time point between the time that the very first foot of the current tripod lands and the last foot of the preceding tripod is lifted (**Figure 4A**). Similarly, we set the tripod end as the halfway point between the very first foot landing time of the following tripod and the last lift-off time of the tripod of interest (dotted blue lines in **Figure 4A** mark the start and end of each tripod). We will refer to the tripod stance as a step.

*Figure 4A* shows the speed profile during a fast step. As previously reported for stick insects (*Graham, 1985*) and *Drosophila* (*Graham, 1972*; *Mendes et al., 2013*), the CoM typically reached a maximum horizontal speed at mid-stance (**Figure 4A**). **Figure 4B** shows the speed for a slower step. A slow step is characterized by both a lower initial speed and a smaller speed increase.

It is important to note that there is a mid-stance maximum in the height of the fly and that the flies are more erect when they are walking faster (**Figure 4A, B**). We will show later that this change in height is partially responsible for the increase in speed.

The change in speed within a step increased with the average speed during the step. (**Figure 4C**). At low walking speed, much of the change in speed was due to the increased initial speed. At higher walking speed, the mid-stance increases in speed made a greater contribution (**Figure 4D**).

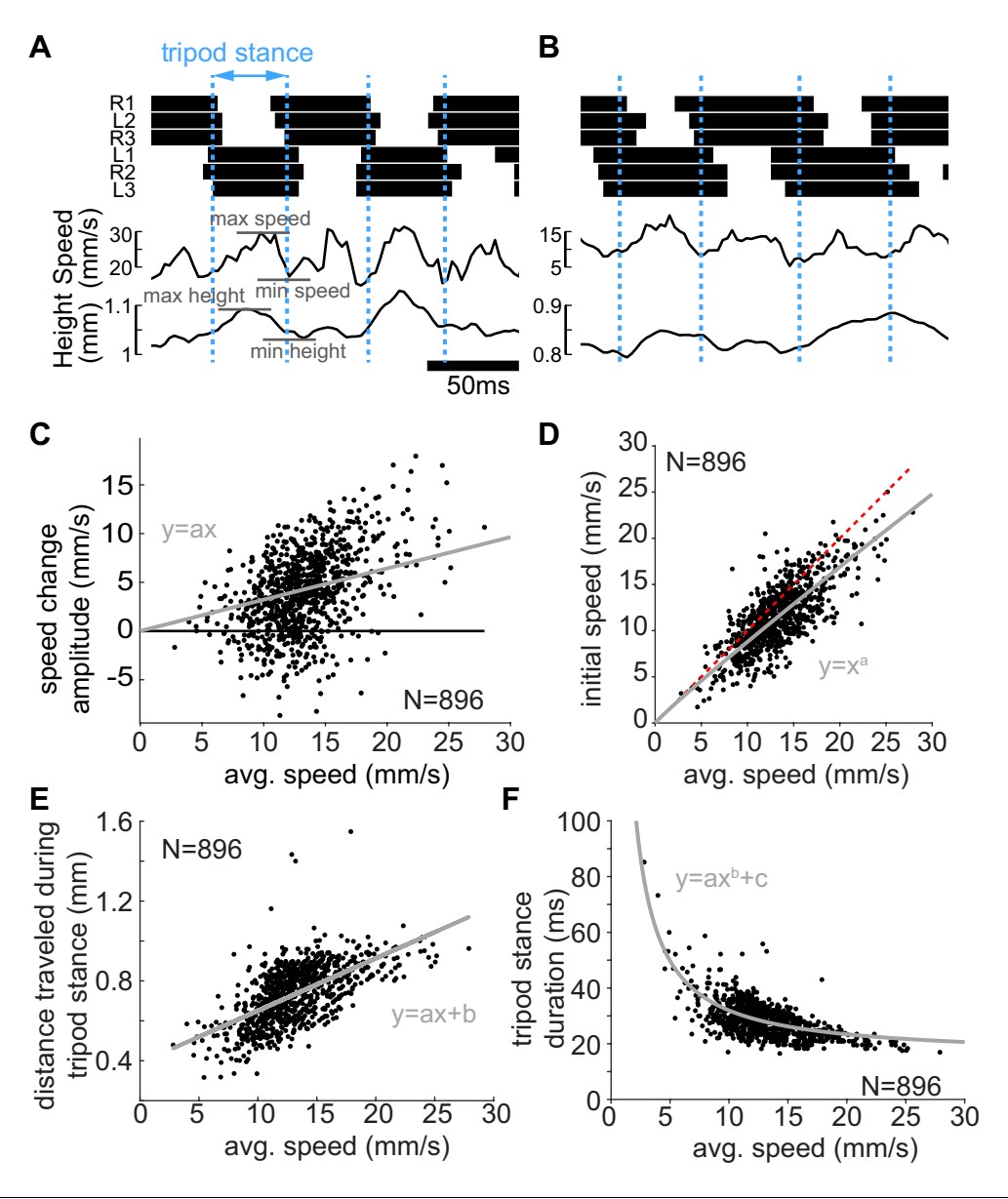

**Figure 4.** Mechanical changes associated with changes in speed. (A) Example trace showing the changes in center of mass (CoM) speed and height. Dotted blue lines are the boundaries between consecutive tripod stance (see text). The CoM shows clear mid-stance maxima in height and speed. (B) A slower step is characterized by smaller initial speed, smaller speed changes, and lower height. (C) Within step speed changes increase with speed (p<0.0001). Line is the best fit of y=ax to the data. a = 0.32. (D) The initial speed increases as the fly walks faster, but this increased speed makes a smaller contribution at higher speed reflected by the increased deviation (gray line is the best fit) from the line of unity (dotted red line). a = 0.9438; 95% confidence bounds (0.9393, 0.9484). (E) Distance traveled during tripod stance increases with speed (p<0.0001). a = 0.026, b = 0.39. (F) Tripod stance duration decreases with speed. a = 187.8, b = −1.056, c = 15.45.

The online version of this article includes the following figure supplement(s) for figure 4:

**Figure supplement 1.** Calculation for height and speed change.

The distance traveled over the tripod gait cycle also increases with speed (*Figure 4E*), and the duration of the tripod gait cycle decreases (*Figure 4F*). Thus, the increase in speed is due to both the greater distance travelled during the step, and faster steps. The longer and faster steps result both from a faster speed at the beginning of the step and a greater increase in speed during the step.

In the rest of the article, we will describe a simple mechanical model that not only describes the mid-stance maximum in speed during a step but also describes the changes in mechanics underlying changes in speed.

## A new mechanical model for locomotion in insects

As described in the 'Introduction', an animal walking with a tripod coordination can be modeled as a springy tripod where a point mass is supported by three legs. For symmetry, in our model, these legs were of equal natural length, and the body's movement within a step can be described as an arc about the middle leg. The body's position at any instant is described by $\theta$, the angle the body makes with the vertical, and $r$ is the length of the middle leg. The behavior of this mechanical system can be described by its elastic potential energy. As the body moves through its stance phase, this elastic potential energy changes as some legs stretch and others compress. The total elastic potential energy of a springy tripod (*Figure 5*) is simply the sum of the potential energies due to the three legs.

$$V_{tri} = \frac{1}{2}k\left(R_{tri} - \sqrt{r^2 + L^2 - 2rL\sin\theta}\right)^2 + \frac{1}{2}k\left(R_{tri} - \sqrt{R_{tri}^2 + L^2 + 2rL\sin\theta}\right)^2 + \frac{1}{2}k(R_{tri} - r)^2 \tag{1}$$

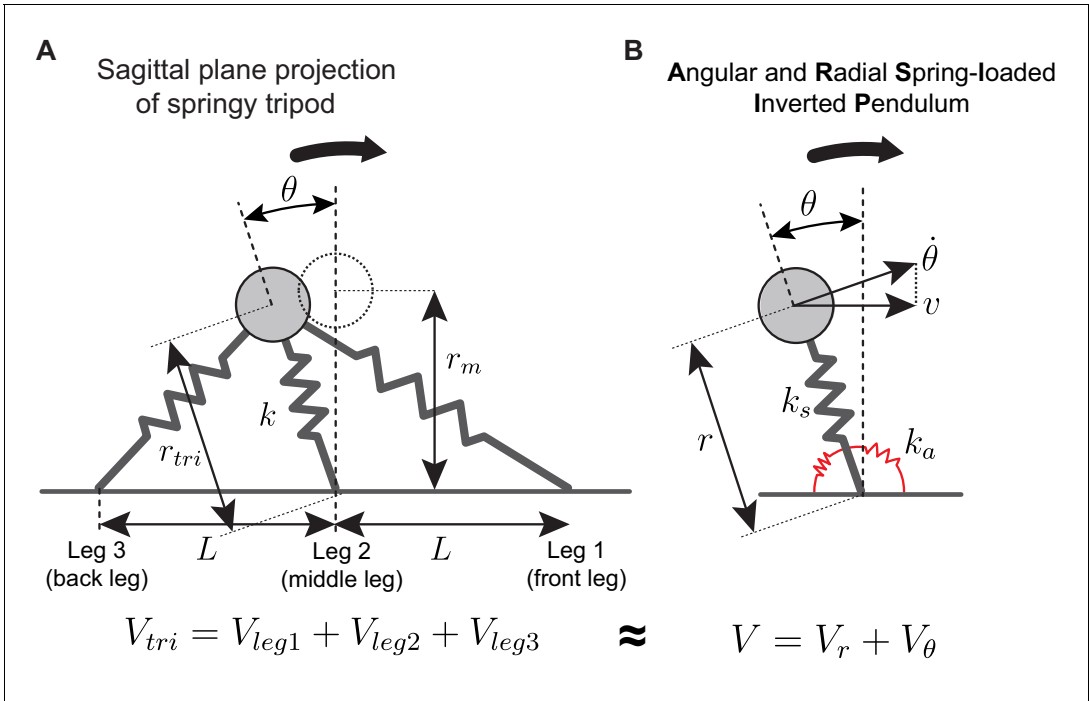

**Figure 5.** The angular and radial spring-loaded inverted pendulum (ARSLIP) model is equivalent to the springy tripod model. (**A**) The sagittal plane mechanics is governed by the sagittal plane projection of the springy tripod. The arrow denotes the direction of motion. The parameters that define the springy tripod model are shown. The overall stiffness of the springy tripod is determined by the spring constant of individual legs, $k$, the height of the tripod ($r_m$), and the distance between the front and back legs (2L). See *Table 1* as well. The behavior of the springy tripod is described by how the coordinates of the point mass – $r$ and $\theta$ – change with time. (**B**) The sagittal plane projection of a springy tripod can be modeled as the ARSLIP model. The parameters that describe the ARSLP model are shown. Just like the springy tripod, ARSLIP is described by how the coordinates of the point mass – $r$ and $\theta$ – change with time. The potential energy of the tripod can be derived as a sum of the elastic energies of the three legs. The ARSLIP potential energy can be derived by summing radial and angular potential energies. The equivalence of the two models is shown by finding parameter set for ARSLIP where the potential energies as a function of $r$ and $\theta$ are similar when $\theta$ is small and changes in $r$ are small (derived in 'Materials and methods').

**Table 1.** Notations used in the article.

| Symbol | Definition and units | Other explanation |
|---|---|---|
| $r$ | Radial coordinate | Variable in ARSLIP and SLIP models |
| $\theta$ | Angular coordinate | Variable in ARSLIP and SLIP models |
| $R$ | (AR)SLIP spring natural length | Optimized for each fly. Set to be within 10% of the measured mesothoracic leg |
| $R_{tri}$ | Tripod spring natural length | Optimized for each fly |
| $k_s$ | Radial spring constant | Variable in ARSLIP and SLIP models |
| $k_a$ | Angular spring constant | Variable in ARSLIP and SLIP models |
| $k$ | Spring constant for individual legs | Variable in springy tripod |
| $r_m$ | Mid-stance height | Experimentally determined for each step (see *Figure 7A*) |
| $v$ | Speed or horizontal velocity | |
| $V$ | ARSLIP potential energy | |
| $\bar{r} \equiv r/R$ | Nondimensional radial coordinate | |
| $\gamma_s$ | Nondimensional radial spring constant | $\gamma_s = \frac{k_s R}{mg}$ |
| $\gamma_a$ | Nondimensional angular spring constant | $\gamma_a = \frac{k_a}{mgR}$ |
| $\bar{r}_m$ | Nondimensional mid-stance height | $r_m / R$ |
| $\Omega$ | Nondimensional mid-stance angular speed | |
| $Fr$ | Froude number – nondimensionalized speed | $v^2/(g*leg\ length)$ |
| $L$ | Tripod spread | Experimentally determined for each step (see *Figure 7A*) |
| $V_{tri}$ | Tripod potential energy | |
| $g$ | Gravitational constant | |
| $m$ | Mass of the fly | Mass was kept fixed; average mass of flies of a particular sex and genotype was measured |
| ARSLIP | Angular and radial spring-loaded inverted pendulum | |

where $R_{tri}$ is the natural length of the springy tripod; $r$ is the length of the middle leg; $\theta$ is the angle that it makes with the vertical axis, which are also identified with the radial and angular coordinate of ARSLIP; $2L$ is the spread of the tripod or the distance between the prothoracic and metathoracic legs of the tripod in the direction of walking; and $k$ is the stiffness of each leg. The variables are also enumerated in *Table 1* and shown in *Figure 5*.

We can show through a formal analysis using a Taylor series expansion of the Lagrangian for the springy tripod (see 'Materials and methods' 'Derivation of the formula relating tripod model to ARSLIP') that springy tripod approximately reduces to the ARSLIP model (*Figure 5B*). As a reminder, in the ARSLIP model, the three legs of the springy tripod are replaced by a single effective leg with a radial and an angular spring. Specifically, $V_{tri}$ is equivalent to the ARSLIP potential energy, $V$ (*Equation 2*), for evolution that is close to the midpoint ($r_m$), and $\theta = 0$.

$$V = \frac{1}{2}k_s(R_{tri} - r)^2 + \frac{1}{2}k_a\theta^2 \qquad (2)$$

The $(r - R)^2$ term corresponds to potential energy due to the radial spring aligned along the effective leg connecting the middle tripod leg to the CoM, and the $\theta^2$ term corresponds to the potential energy from an angular spring capturing the tangential restorative forces exerted by the front and back legs. In this model, the mechanics of an animal is controlled by the two spring constants, $k_s$ and $k_a$, which describe the stiffness of the radial and the angular spring, respectively, and the natural spring length, $R$.

In essence, the mechanics of the fly walking on a springy tripod can be described by the ARSLIP model. An important point that we will elaborate on later is that the springy tripod is a simplification of the actual configuration of the fly while it is walking, but the ARSLIP model is a more general

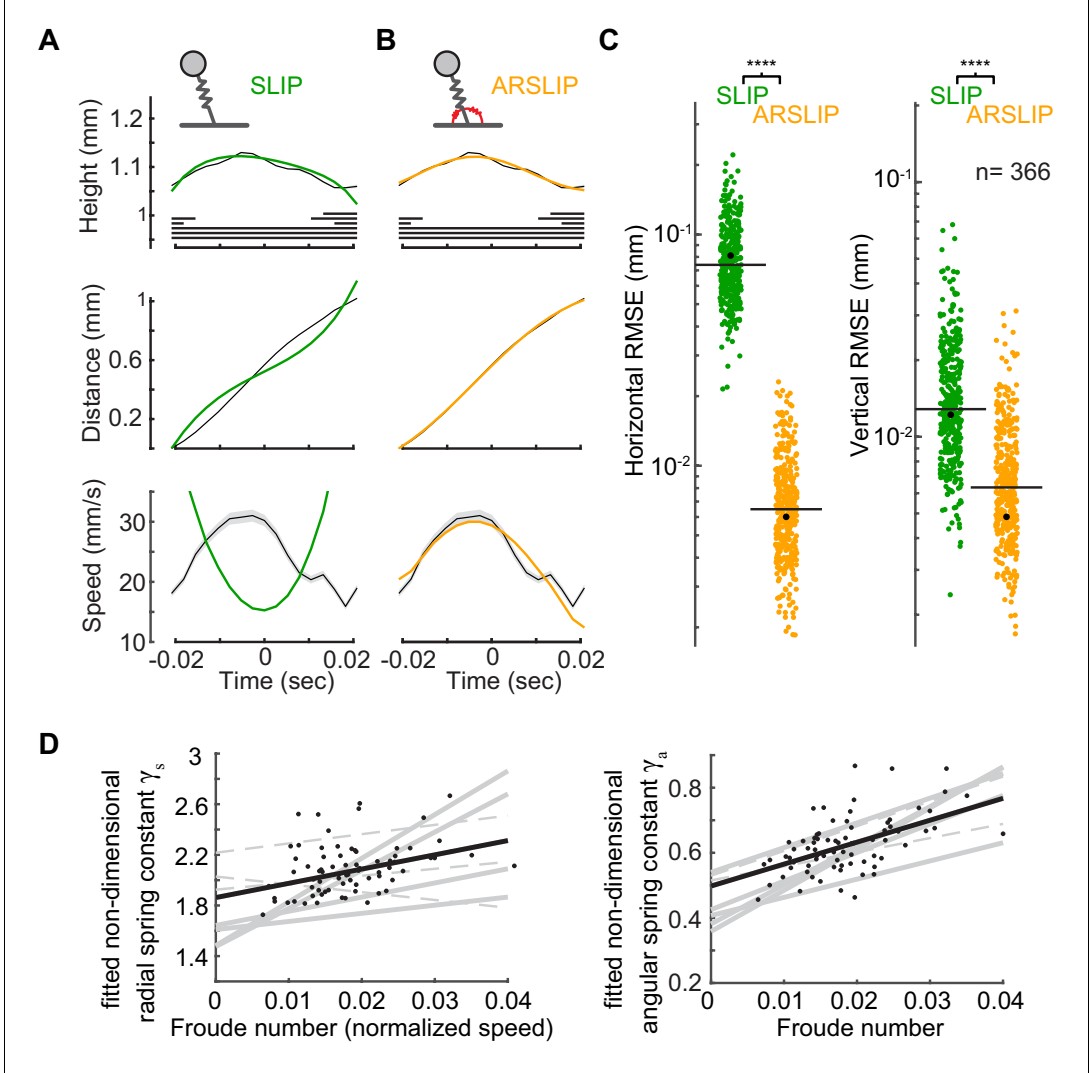

**Figure 6.** Angular and radial spring-loaded inverted pendulum (ARSLIP) describes the kinematics of a fly's center of mass (CoM). (**A**) The best fit of SLIP (green) and (**B**) ARSLIP models – to the height (top) and horizontal position of the CoM (middle). Speed (bottom) was not used to fit but is plotted to simply show why SLIP fails. Gait maps are shown with CoM height. (**C**) ARSLIP is a significantly better model than SLIP (Wilcoxon rank-sum test; p<0.001 for both horizontal and vertical movement). Each dot is the root mean squared error (RMSE) for the best fit to a single tripod stance. The example step presented in (**A**) and (**B**) is marked as a black dot and was chosen close to the SLIP median RMSE. Black horizontal line is the median. (**D**) $\gamma_s$ (left) and $\gamma_a$ (right), which are nondimensionalized spring constants, increase with speed in most flies. Each line is a fit to the steps from a single fly. Black dots (n = 74) show values corresponding to individual steps for one fly and the fit corresponding to that fly. Solid lines are the regressions with p<0.05 from F-test, and dotted lines are p>0.05.

The online version of this article includes the following figure supplement(s) for figure 6:

**Figure supplement 1.** Fit parameters of the angular and radial spring-loaded inverted pendulum model.

model and can serve as an accurate model even without making the assumptions of the springy tripod.

The approach above is based on potential energy. The distinction between SLIP and ARSLIP is clearer when considering forces modeled in the two cases. In SLIP, forces are always along the leg. The ARSLIP model provides a mechanism by which tangential forces can also be transmitted to the body. Importantly, the angular spring forces switch direction at mid-stance, which means that they aid forward progression during the first half of the stance and oppose forward progression during the second half of the stance. This pattern is exactly opposite to the pattern created by SLIP.

Depending on whether the leg spring dominates, or the angular spring, one can get a cockroach-like speed minimum or fly-like speed maximum at mid-stance.

## ARSLIP models the kinematics of a fly's CoM during walking

We evaluated the performance of the SLIP and ARSLIP models by fitting them to the fly's CoM kinematics. Because the stance times of two consecutive tripods can overlap substantially, a complete model would involve two effective legs, each of which functions as either SLIP or ARSLIP; this complete model with two effective legs would have too many parameters and might obscure many of the insights that we obtain from modeling. Therefore, we modeled the CoM kinematics of a tripod stance (as defined in *Figure 4A*) using a single effective leg. The model parameters approximate the control exerted by the fly at each step. In the ARSLIP model, the fly chooses as its initial condition the angle of attack ($\alpha$), angular speed ($\Omega$), leg length ($r$), and radial speed ($\dot{r}$) at the beginning of the step. The evolution of the CoM depends on the angular spring constant ($k_a$), leg spring constant ($k_s$), and the natural leg length of the effective leg ($R$). The only difference between ARSLIP and SLIP is the absence of the angular spring, and hence there is no $k_a$ in SLIP.

We minimized the root mean squared error (RMSE) between the SLIP and ARSLIP-predicted position of the CoM, and the experimentally measured position using an optimization algorithm (see 'Materials and methods', 'System of ordinary differential equations for SLIP and ARSLIP and details regarding fitting ordinary differential equations to individual steps'). SLIP can model the small increase in the vertical position of the CoM, which results from two competing effects: height increase due to the progression of the CoM from its extremum to the vertical mid-stance position and a height decrease due to the compression of the leg spring (*Figure 6A*). However, as reported previously (*Biswas et al., 2018*), SLIP fails to describe the horizontal progression of the CoM. This failure is clear from a comparison of the experimental horizontal speed profile and the theoretical speed profile (*Figure 6A*, bottom panel). In contrast, ARSLIP can describe both the horizontal and vertical progression of the CoM (*Figure 6B*). In ARSLIP, the angular spring accelerates the CoM during the first half of the stance phase. It can, therefore, compensate for, or even overcome, the decelerating effects of the radial spring and gravity, and can model the mid-stance maximum in speed. ARSLIP presented significantly smaller RMSEs for both horizontal and vertical CoM displacements than SLIP (*Figure 6C*).

That the ARSLIP model describes the CoM kinematics well means that two linear springs defined by their spring constants – $k_a$ and $k_s$– are sufficient to describe the fly's CoM kinematics during a step. The range of parameter values for all the fitted steps in our dataset is shown in *Figure 6—figure supplement 1*. The median $k_s$ was 0.009 N/m. This spring constant implies that to support its mass of 1 mg or 10 µN weight, the fly compresses this effective spring by about 1 mm or approximately 50% of its length. During a step, the spring is always compressed such that its length is close to the fixed point of the spring (length at which the spring forces cancel gravitational forces) and oscillates about this fixed point without reaching its natural length. The magnitude of these oscillations is small and reaches a maximum of 10% (of its length at fixed point) about the fixed point. The nondimensional radial spring constant $\gamma_s$ (see 'Materials and methods' for definition) is ~2 compared to >10 for humans (*Antoniak et al., 2019*). The median $k_a$ was $1.1 \times 10^{-8}$ Nm/radian. In nondimensional terms, the angular spring constant $\gamma_a$ was ~0.5, which is like the values obtained in humans. Therefore, compared to humans, the relative role of angular spring in flies is much larger.

Most animals increase their walking speed by decreasing the stance duration (*Mendes et al., 2013*; *Pearson, 1976*). Modeling stance using an effective limb that functions as a spring provides a simple explanation for the mechanical changes that accompany this decrease: in any two-dimensional motion – including walking in the sagittal plane discussed here – the vertical and horizontal motion must be synchronized by relating parameters governing the vertical and horizontal time scale. The vertical oscillatory motion is controlled by the radial spring constant, and the horizontal motion by the angular speed. As the walking speed increases and the stance duration decreases, the vertical oscillations must occur faster by making the effective leg stiffer (because the time needed for vertical oscillation decreases with increases in spring stiffness). This increase in stiffness has indeed been observed in humans (*Antoniak et al., 2019*). We found a similar increase in stiffness in flies (*Figure 6D*). As the fly's walking speed increases, $\gamma_s$, the nondimensionalized version of $k_s$

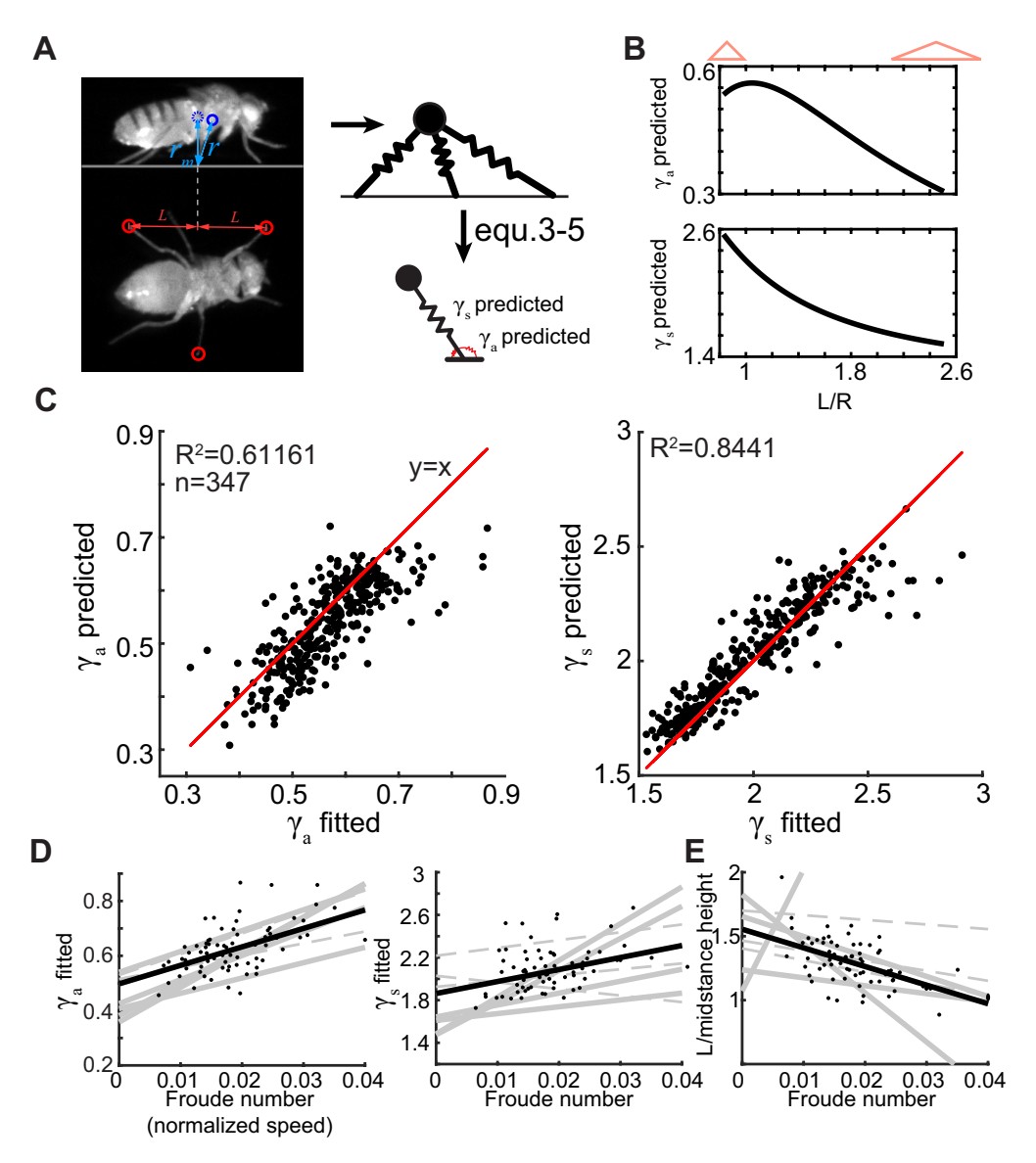

**Figure 7.** Flies increase their height and decrease the extent of their tripod as they increase their speed. (A) We measure $r_m$ and L for each step which describe the tripod geometry during a step. The tripod geometry can be used to predict $\gamma_a$ and $\gamma_s$ using *Equations 3–5*. (B) $\gamma_a$ (top) and $\gamma_s$ (bottom) decrease as $L/r_m$ increases. Narrow and high tripods are stiffer (see red cartoons atop). (C) The spring constants predicted from geometry closely correlate to the optimal spring constants calculated in *Figure 6* ($\gamma_a$: left and $\gamma_s$: right). The red line is the line of unity. (D) $\gamma_a$ (left) and $\gamma_s$ (right) replotted from *Figure 6* show increase with speed. (E) Increases in Fr in most flies are concomitant with decrease in $L/r_m$. Each line is a fit to a different fly. Solid lines are the regressions with $p<0.05$ from F-test, and dotted lines are $p>0.05$. Black dots (n = 74) show values corresponding to individual steps for one fly, and black line is a fit for that fly.

increases. The nondimensional angular spring constant, $\gamma_a$, increases as well; this increase accounts for the greater within-step increases in speed observed at higher speed (see *Figure 4C*).

In the next section, we will show that the mechanism underlying the change in spring constant is surprisingly a result of the change in the geometry of the tripod with speed rather than the change in spring constant of the individual legs.

## Change in tripod geometry increases spring stiffness necessary for change in speed

In bipedal walkers, the only mechanism for increasing the overall stiffness of the system is to increase the stiffness of each leg. In the case of polypedal walkers such as insects, including flies, the geometry of the tripod itself is a parameter that can be adjusted to alter the stiffness of the overall system. To test the extent to which the geometry of the tripod on a given step influences its kinematics on that step, we derived the equivalent ARSLIP model that displays the same dynamics as the springy tripod around its mid-stance position using the empirically obtained tripod geometry (determined by the tripod spread, $L$, and mid-stance height, $r_m$, in *Figure 7A*) and mass of the fly (see 'Materials and methods' for details). Specifically, using the following equations:

$$k_s(R - r_m) = k\left[R_{\text{tri}} - r_m\left(3 - \frac{2R_{\text{tri}}}{\sqrt{L^2 + r_m^2}}\right)\right] \tag{3}$$

$$k_a = \frac{2kL^2 r_m^2 R_{\text{tri}}}{\left(L^2 + r_m^2\right)^{3/2}} \tag{4}$$

$$k_s = k\left[3 - \frac{2L^2 R_{\text{tri}}}{\left(L^2 + r_m^2\right)^{\frac{3}{2}}}\right] \tag{5}$$

We can relate the ARSLIP spring constants – $k_a$ and $k_s$ and R (the natural length of the ARSLIP spring) to $k$, the spring constant of individual legs, $R_{tri}$, the natural length of individual legs, and the tripod geometry (determined by $L$ and $r_m$ in *Figure 7A*).

If the spring constant of individual legs remains the same, the springy tripod model predicts that $\gamma_a$ and $\gamma_s$ (nondimensionalized versions of $k_a$ and $k_s$) will both vary over a twofold range due to the variation in the observed geometry of the tripod. Specifically, in the range of values observed in flies, as $L/r_m$ ratio decreases or the tripod becomes narrow and tall, both $\gamma_a$ and $\gamma_s$ increase (*Figure 7B*).

We can exploit this dependence of $\gamma_a$ and $\gamma_s$ on the tripod geometry to examine how well the change in tripod geometry from one step to the next predicts the best fit $\gamma_a$ and $\gamma_s$ values from *Figure 6*. To this end, we determined a single $k$ and $R_{tri}$ for each fly, which best satisfies *Equations 3–5* for all the steps fitted with the ARSLIP model for that fly. To compare across flies, we converted the $k_a$ and $k_s$ values to their nondimensionalized versions $\gamma_a$ and $\gamma_s$. We found that, despite all the simplifying assumptions, the predicted $\gamma_a$ and $\gamma_s$ derived from the tripod geometry were close to the optimal $\gamma_a$ and $\gamma_s$ obtained from the best fit to the CoM kinematics (*Figure 7C*). The similarity between predicted and fitted spring constants is particularly significant because the prediction for all the steps of a given fly was made with a single parameter set while fits were optimized for each step representing a large decrease in the number of parameters. These results show that the tripod geometry plays a critical role in governing the spring constants.

The strong correlation between spring constants predicted from the geometry and those from optimization suggests that changes in tripod geometry are employed by the fly to change speed. Since $\gamma_a$ and $\gamma_s$ increase with speed (replotted in *Figure 7D*), we anticipate that the increase in speed is usually reflected as a change in $L/r_m$ ratio (*Figure 7E*), implying that the change in tripod geometry is an important mechanism for the control of speed during walking.

## Discussion

There are four main findings in this study:

- Flies use a M-tripod gait across all speeds.
- Faster steps are accompanied by higher initial speed and larger increases in speed during the step, resulting in the fly covering longer distances in a shorter time. The kinematic pattern during a step and its changes with speed are explained by a new model – ARSLIP – within which the dynamics are described by two spring constants. An increase in speed is accompanied by an increase in the spring constants that characterize the ARSLIP model. This increase in stiffness is an important biomechanical adaptation necessary for change in speed.

- The increased stiffness is not a result of each leg becoming stiff but results from a change in the geometry of the tripod and the height of the fly: flies locomote with a narrower and higher posture, resulting in increased stiffness at a higher speed.
- To our knowledge, the effect of tripod geometry on insect locomotion has not been investigated. Because the tripod geometry varies widely between insects, the tripod geometry might be an important determinant of an insect's walking kinematics and has the potential to explain many features of insect locomotion (see last section of 'Discussion') including the fact that insects can walk with a tripod gait across a large range of speeds.
  These findings are discussed below.

## Flies employ a tripod coordination during forward walking

Flies appear to predominantly employ a gait close to a tripod gait – M-tripod – across their entire range of speeds. These results are consistent with other studies in flies (*DeAngelis et al., 2019*; *Strauss and Heisenberg, 1990*). Similar observations have been made during free walking in other insects such as cockroaches (*Delcomyn, 1985*; *Hughes, 1952*; *Spirito and Mushrush, 1979*), ants (*Reinhardt and Blickhan, 2014*; *Wahl et al., 2015*; *Zollikofer, 1994*), and locusts (*Burns, 1973*).

The M-tripod gait itself is not fixed but has a small dependence on speed. This small dependence on speed as well as the increase in duty factor as the speed decreases implies that the average number of legs on the ground at any given time will decrease with speed. This increase in the number of legs has been shown to be important for stability (*Szczecinski et al., 2018*).

The M-tripod gait only applies to forward walking at a fixed speed. The entire complement of gaits that the fly employs to turn, accelerate, and decelerate remains to be determined. It is important to note too that our findings do not imply that the flies are only capable of a fixed gait. There is evidence that flies change their gait upon amputation (*Isakov et al., 2016*). In our dataset as well, there is clear evidence for tetrapod gait; however, the fraction of steps during which flies adopt a tetrapod gait is very small.

## The geometry of the tripod is an important determinant of walking speed

An unexpected result was the extent to which the shape of the tripod formed by the three tripod legs – particularly the ratio of the height of the tripod to its anterior-posterior spread – can explain a fly's CoM kinematics during a step. Previous studies have shown that neither the swing duration nor the swing amplitude (the distance that a leg travels during the swing) changes much as the fly's walking speed changes (*Mendes et al., 2013*; *Strauss and Heisenberg, 1990*; *Wosnitza et al., 2013*), a result that is confirmed in this study. Much of the change in speed results from a decrease in stance duration (*Mendes et al., 2013*; *Strauss and Heisenberg, 1990*; *Wosnitza et al., 2013*), another result consistent with this study. In other words, increase in walking speed results from an increase in the angular speed of the body about its stance legs. This increase requires two elements: a neural element whereby increasing the drive into the central pattern generators would cause them to cycle faster, as has been demonstrated in stick insects (*Büschges et al., 2008*). A biomechanical element: moving faster also requires larger forces from the ground, and a mechanically stiffer system (in this case, the mechanical system consists of the fly and the legs that support the fly) would be able to transmit more forces from the ground to the body. There are two mechanisms by which the system can become mechanically stiffer. Either a fly could make each leg stiffer just as has been shown in humans (*Antoniak et al., 2019*; *Kim and Park, 2011*) or it could change the geometry of the tripod to make the overall system stiffer. The data in *Figure 7* is consistent with the second idea that the changes in the geometry of the tripod are the dominant component by which flies control the stiffness of $\gamma_a$ and $\gamma_s$ and thereby change their walking speed. Changes in $\gamma_s$ allow the fly to adjust the stiffness of its mechanical system to the stance duration; a stiffer $\gamma_s$ means shorter time period of oscillation. Changing $\gamma_s$ through changes in geometry would also change $\gamma_a$. We regard the changes in $\gamma_a$ as an inevitable consequence of the changes in $\gamma_s$; nonetheless, the increase does provide a parsimonious explanation for the greater mid-stance maximum in speed observed when the fly walks faster.

To our knowledge, the control of speed through tripod geometry has never been explored in any insect. One reason for this deficiency is methodological. Researchers usually collate their data across steps, trials, and individuals. This process is bound to obscure any trends in tripod shape; analysis at

the level of single steps is necessary, and trends within an individual must be compared. Another methodological issue is that the height of the animal during locomotion is rarely measured.

### The large variation in tripod geometry can explain the broad range of kinematics observed among insects

In flies, the $L/r_m$ ratio varies between 1 and 2. How about other insects? We could obtain a rough estimation of the $L/r_m$ ratio for a few insects by piecing together information from some manuscripts or by measuring these ratios from the figures in the papers: for three species of stick insects, the ratio ranges between 2 and 3 (*Theunissen et al., 2015*), wood ants have a $L/r_m$ ratio of closer to 1 (*Reinhardt and Blickhan, 2014*), and cockroaches have a $L/r_m$ ratio closer to 6 (*Ting et al., 1994*). This large variation in the $L/r_m$ ratio will have a large effect on the CoM kinematics within a step. Given a leg stiffness γ, at low $L/r_m$ ratio, that is, when the legs are almost vertical, $γ_s$ is large in comparison to $γ_a$; therefore, the deceleration due to $γ_s$ as the fly approaches mid-stance cannot be compensated by the acceleration due to $γ_a$. As the $L/r_m$ ratio increases, the effects due to $γ_a$ and $γ_s$ are comparable for a range of $L/r_m$ values. At very large $L/r_m$ ratios, $γ_s$ again dominates. These ideas can be formalized by deriving conditions for which the gait is cockroach-like versus fly-like (Appendix): whether the gait is cockroach-like or fly-like depends on the interplay between $γ_a$ and $γ_s$, which in turn depends on the stiffness of individual legs and the geometry of the tripod. For a given $L/r_m$, there is a leg stiffness γ above which the kinematics change from a cockroach-like gait to a fly-like gait (*Figure 8A*). As expected, the data points for each of the fitted steps for the fly lie above the function that demarcates the two kinematic types. Importantly, the $L/r_m$ ratio for flies and several

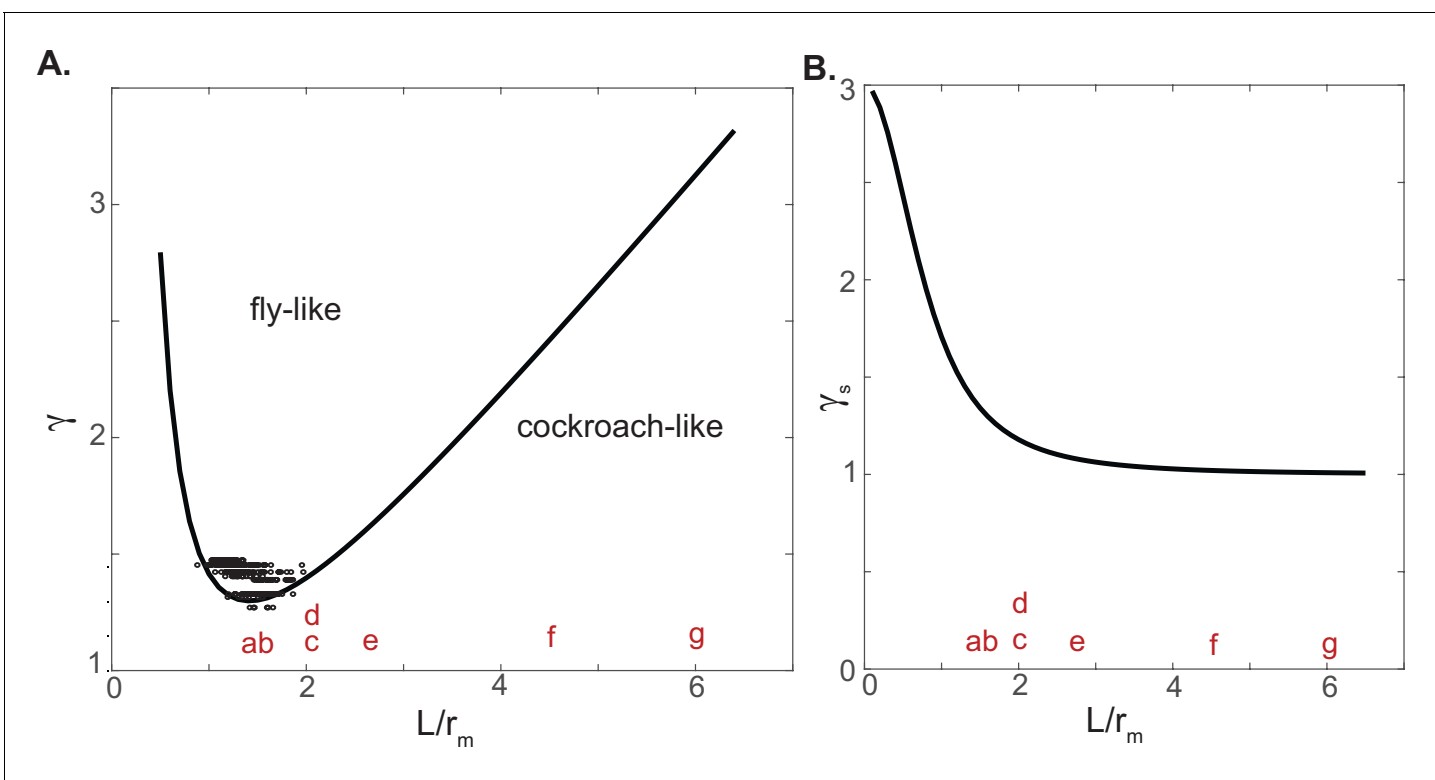

**Figure 8.** The large variation in tripod geometry can explain why insects that employ a tripod gait have varying speed and kinematics. Two parameters that affect the stiffness of an animal walking on a tripod are the nondimensionalized stiffness of each leg (γ) and the geometry of the tripod defined by the ratio L/r_m. (A) The curve divides the γ-L/r_m plane into two regions with either fly-like or cockroach-like kinematics. The data points show the values from fits to the fly data. The alphabets represent approximate L/r_m ratio for a few insects as noted below. Most insects have an L/r_m ratio near the minimum of the curve, making it more likely that they can achieve an inverted fly-like kinematic profile. (B) Plot of leg spring constant as a function of L/r_m ratio. Many insects have L/r_m ratio such that changes in L/r_m ratio can affect γ_s. Cockroaches have L/r_m ratio where changes in geometry would have little effect on the spring constant.   a=Formica polyctena; b=Formica pratensis; c=Carausius morosus; d=Aretaon asperrimus; e=Medauroidea extradentata; f=Periplaneta americana; g=Blaberus discoidalis.

other insects places them in a regime in which the leg stiffness required for a fly-like gait is at a minimum (*Figure 8A*). Indeed, ants, stick insects, and flies all have fly-like kinematics. On the other hand, cockroaches have $L/r_m$ values that predispose them toward a mid-stance minimum velocity profile.

Another important insight from this analysis comes from the dependence of $\gamma_s$ on $L/r_m$ ratio (*Figure 8B*). At the $L/r_m$ ratios observed in flies and other insects such as ants and stick insects, small changes in $L/r_m$ will produce a corresponding change in $\gamma_s$ (*Figure 8B*). On the other hand, at large $L/r_m$ ratio, such as those employed by a cockroach, the $\gamma_s$ is constant and does not depend on the $L/r_m$ ratio. This difference makes sense. At the large *Fr* numbers employed by cockroaches, the CoM kinematics is dominated by the angular speed with which the body moves about its leg (*Antoniak et al., 2019*), and it is rather insensitive to the magnitude of $\gamma_s$. On the other hand, and as demonstrated for flies in this study, at lower speeds the mechanics is dominated by the spring constants $\gamma_a$ and $\gamma_s$.

In sum, the strong dependence of mechanics on the $L/r_m$ ratio or tripod geometry can explain both why tripod gait can be employed across a range of speeds observed among different insects and that a given insect can control its speed in part by changing the geometry of its tripod.

## ARSLIP as a general model for multilegged locomotion

The finding that the SLIP model employed in mammalian locomotion is also adequate as a model for cockroach running led to the idea that SLIP is a general model for locomotion regardless of how many legs are on the ground. However, it has been well known before the recent studies in flies that CoM kinematics for other insects are dramatically different from cockroach (*Graham, 1985*; *Reinhardt and Blickhan, 2014*), and cannot be explained by the SLIP model. To our knowledge, there have not been many attempts at arriving at a single model that can serve as a conceptual model that explains the diverse kinematics observed across the insect world.

The development of a general model will be aided by two important insights in this study: first, we show that the geometry of the tripod plays an important role in determining the mechanics of the CoM. It appears that cockroaches do walk with a particularly wide and low tripod where the mechanics are closest to being described by the SLIP model. But the wide and low tripod observed for cockroaches appear to be an exception rather than a rule. The tripod geometries of other insects are different, and as shown in *Figure 8*, at least some of the kinematic variations observed in insects result from these differing tripod geometries.

Second, the effect of the different tripod geometries can be captured through a simple extension of the SLIP model – the ARSLIP model. The ARSLIP model is simple enough that it retains much of the elegant simplicity that makes SLIP such a powerful model while being better equipped to capture the greater diversity of kinematics observed in insects.

Future experiments aimed at exploiting the natural diversity of the insect world to rigorously test the relationship between the geometry of the stance legs and CoM kinematics, and the ability of ARSLIP to describe this relationship will prove to be a powerful toolkit for developing a general model for hexapedal locomotion.

It is obvious but still important to note that the springy tripod presented here is a simplification for the actual dynamics of locomotion in insects. In the case of fly, the springy tripod is a decent model of the fly's walking as shown by its ability to predict the optimized ARSLIP spring constants from geometry (*Figure 7*). However, the ARSLIP model itself is more general and can model other known features of insect legged locomotion that we have not considered here:

First, here we have modeled each leg as a linear spring, the angular spring results from a combined action of the three legs. In the most general case, each leg itself can function as both linear and angular spring. The resulting model will still be the ARSLIP model; however, the expressions relating $k_a$ and $k_s$ to tripod geometry and stiffness of a given leg will be different from what we have derived here.

Second, and like the first point above, apart from spring forces, insect legs can produce attachment forces (*Gorb et al., 2002*). Once again, attachment forces do not invalidate the utility of ARSLIP as a model, but will affect the values of $k_a$ and $k_s$ differentially and represent an important mechanism that can explain the difference in kinematics for different insects.

Third, for many insects, the hind leg is much longer than the other legs. The longer length of the hind legs might make a third term in the Taylor series expansion necessary (the first two

terms being the leg spring and the angular spring, respectively). This third term may act as an asymmetric propulsive force.

Finally, ARSLIP does not necessarily need three legs. The ARSLIP model can also model an insect employing more legs on the ground. In fact, one important insight in this study is that whenever there is more than one leg on the ground, SLIP is unlikely to work as a model. This is because SLIP assumes that the net forces on the CoM act along the single effective leg. When there are more than one leg of the ground, this constraint – forces only along the leg – severely limits the ability of a model to describe locomotion. ARSLIP removes this constraint and allows the description of forces perpendicular to the leg. Thus, the ARSLIP model is the more natural take-off point for efforts to obtain a truly general model for locomotion not only in insects but in multilegged animals in general.

## Materials and methods

### Flies

The flies were reared at 25℃, and 12 hr:12 hr light:dark cycle. Ten minutes before the experiment the flies were removed from a vial and placed under $CO_2$ anesthesia , and their wings were detached using forceps.

We experimented with different wild-type strains to record steps at a range of walking speeds and to ensure that any general principle we discover is indeed general (at least across a range of inbred strains). These wild-type strains were *w1118*, *Berlin K*, and *Oregon-R-C* (or Oregon C) (Bloomington stock numbers: 5905, 8522, and 5, respectively). *Table 2* shows all of the flies in our dataset and the data each fly contributed to the analyses in each figure.

**Table 2.** Number of data points in the plots.

| | | Figures | | | | | | |
| | | *Figure 2D* | | *Figure 3A* | *Figure 3B,E* | *Figure 4C–F* | *Figure 6C* | *Figures 6D, 7C,D* |
| Fly | Genotype | Stance | Swing | | | | | |
|---|---|---|---|---|---|---|---|---|
| 1 | Berlin K | 8 | 9 | 1 | 0 | 2 | 2 | 0 |
| 2 | Berlin K | 4 | 9 | 0 | 0 | 0 | 0 | 0 |
| 3 | Berlin K | 152 | 194 | 20 | 15 | 42 | 14 | 6 |
| 4 | Berlin K | 284 | 330 | 40 | 46 | 82 | 78 | 78 |
| 5 | Berlin K | 72 | 90 | 10 | 12 | 21 | 18 | 18 |
| 6 | Berlin K | 108 | 114 | 16 | 17 | 31 | 28 | 28 |
| 7 | Berlin K | 367 | 458 | 43 | 53 | 97 | 74 | 74 |
| 8 | Berlin K | 388 | 454 | 47 | 59 | 102 | 44 | 44 |
| 9 | Berlin K | 50 | 63 | 5 | 7 | 9 | 0 | 0 |
| 10 | W1118 | 1496 | 1564 | 202 | 227 | 390 | 90 | 90 |
| 11 | Oregon RC | 45 | 53 | 5 | 6 | 5 | 0 | 0 |
| 12 | Tac 3 | 265 | 325 | 32 | 42 | 63 | 5 | 0 |
| 13 | Berlin K | 165 | 209 | 17 | 26 | 38 | 9 | 9 |
| 14 | Berlin K | 53 | 63 | 5 | 9 | 14 | 4 | 0 |
| | Total | 3457 | 3935 | 443 | 519 | 896 | 366 | 347 |

Data used for **Figure 2D** were stance and swing durations for each of the six legs. Data used for **Figure 3A,B and E** were derived from all complete gait cycles that include at least one frame of the last leg's stance phase. Data for **Figure 3A** had an additional constraint that required a cycle data to have complete observation of the last leg's stance phase from start to end. Data used for **Figure 4** were derived from all tripod stance phase. Data used for **Figure 6** and **7** were derived from tripod stance phases in which the single support phase constitutes at least 25% of stance. However, additional constraints were added to **Figure 6D** and **7**: data with erroneous leg position tracks were eliminated, and then flies with less than six steps were eliminated.

## Data acquisition and processing

Our experimental data consisted of the CoM position of the fly in all three dimensions and the position of the fly's footholds in the horizontal plane. This section describes the acquisition and processing procedures that yield this dataset.

### Recording chamber

The chamber side walls and ceiling (inner $L \times W \times H$: $21 \times 7 \times 17$ mm) were built from microscope slides and held together using an instant adhesive (Loctite 495). A hole was drilled in one of the side walls 10 mm from the floor to provide an air jet nozzle for the initiation of walking. A 0.13–0.17-mm-thick coverslip was used as the chamber floor to minimize the distance between the side view and bottom view, and therefore the frame size; the frame size was kept to a minimum to increase the frame rate. After a fly was placed inside the chamber, the chamber was secured on the coverslip using a tape. The chamber–coverslip assembly was then held horizontally using clamps. Below the assembly a mirror was tilted at 45° to the coverslip. The mirror reflected the bottom view of the chamber to the camera (see *Figure 2* for schematic). The bottom and the side of the chamber were lit with infrared light.

### Data acquisition

Data acquisition and processing were fully automated, except for manual screening of raw videos before the processing step. A USB 3.0 camera Basler acA1920-150 um (380 Hz at 1024 × 779) and a telecentric lens (Edmund Optics, Barrington, NJ 0.40x SilverTL, part number 56–677) were used to record the video at 380 fps at 1024 × 779 resolution. Exposure was set at 2.5 ms. This setup had a modulation transfer function of 10% in the vertical direction and 6% in the horizontal direction at 25.39 line pairs/mm. The camera monitored the chamber at 30 Hz in real time until any motion within the field of view triggered acquisition at 380 Hz for 1.2 s. The motion was detected by measuring the change in intensity between the total pixel intensity values of the two most recent frames. After each acquisition, the recorded video was saved to disk if and only if the fly walked more than 5 mm across the floor. This automated procedure could monitor and record a single fly for more than 10 hr.

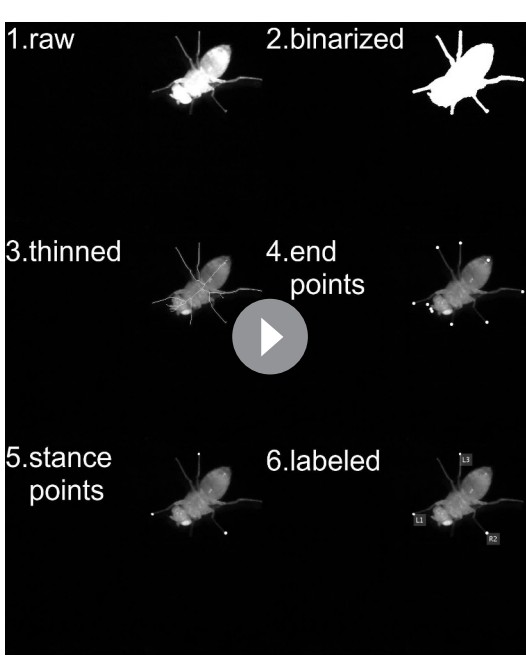

**Video 2.** The output of each processing step is plotted.
https://elifesciences.org/articles/65878#video2

### Tracking CoM and foothold positions

The fly's CoM was estimated by using the most prominent features of the fly as fiducial markers. The features were extracted on the first frame by using the minimum eigenvalue algorithm and following the extracted points throughout the video using KLT (KLT feature tracker in MATLAB [*Lucas and Kanade, 1981*; *Shi and Tomasi, 1993*; *Tomasi and Kanade, 1991*]). An estimated affine transformation matrix between the sets of feature points of consecutive frames was multiplied to the CoM position in the previous frame to evaluate the CoM position in the current frame (*Figure 2—figure supplement 1*). Next, between every pair of consecutive frames, CoM was backtracked one step. The distance between original and backtracked CoMs is a reliable measurement of the so-called forward–backward tracking error (*Kalal et al., 2010*). The error was small enough that we could evaluate the SLIP and ARSLIP models. The errors were also much smaller than the change in CoM position during a step (*Figure 2—figure supplement 2*). Therefore, the noise of the estimated

CoM trajectories was small, so numerical derivations of the trajectories returned velocity trends with a small noise (*Figure 2—figure supplement 2*).

Foothold location was automatically detected using a series of image processing algorithms detailed in *Figure 2—figure supplement 1*. The basic idea was to binarize the bottom view and thin the resulting image to yield a skeleton. The end points of the resulting skeleton returned points, including the actual footholds, along with other noisy or random points. The actual footholds were robustly detected by filtering out the noisy points and extracting points that are located the furthest away from the CoM. The noise filtering was performed by removing small objects composed of fewer than 100 pixels. The legs were labeled based on the mean of each foothold trajectory in the CoM frame (see details in *Figure 2—figure supplement 1*).

All the processing steps are shown in *Video 2*.

## Gait analysis

We performed gait analysis using either the stance start times or the instantaneous phase.

### Gait analysis based on stance start times

For quantifying gait based on stance start times, the time at which the R1 leg starts its stance denoted the beginning of the cycle. The cycle lasted until the R1 leg entered the next stance. The time between R1 entering two consecutive stances was the cycle period. The time delay between R1 and other legs was calculated by taking the time difference between the stance times of the other legs, allowing for the fact that some of the legs would start their stance before the R1 leg. To normalize the time delays, these delays were divided by the cycle period.

### Gait analysis based on leg phases

The position of legs, $y_i(t)$, was measured in a body coordinate system, and the positive *y*-axis points toward the anterior part of the body. Because we only knew leg position during stance, we performed a linear interpolation of $y_i(t)$ during swing. The instantaneous phase angles of the legs were obtained by applying Hilbert transform on $y_i(t)$ (*Figure 3*; *Couzin-Fuchs et al., 2015*; *Revzen and Guckenheimer, 2008*; *Wilshin et al., 2017*). Hilbert transform turns a real-valued signal into a complex-valued analytic signal, which provides accurate instantaneous magnitude and frequency of the real-valued signal (*Boashash, 1992*; *Marple, 1999*; *Smith, 2007*). The time-dependent angle of complex-valued analytical signal is the instantaneous phase angle. The phases were between [$-\pi$, $+\pi$] (see *Figure 3D* for definition). For the phase delay analyses (*Figure 3D–F*), the phase delay was normalized to [$-0.5$, $0.5$] by dividing the instantaneous delays by $2\pi$. This normalized phase delay for each leg relative to R1 was averaged over the entire stance phase of the R1 leg (touch-down to lift-off).

### Definition of M-tripod

Median values of delays between mesothoracic and contralateral metathoracic legs $\Delta_{meta,meso}$ and delays between prothoracic and contralateral mesothoracic legs $\Delta_{meso,pro}$ are used to determine delays within tripod legs of a synthetic M-tripod.

## The calculation for change in height and velocity

A time series of height or speed over a tripod stance was detrended by a line that connected the values at the beginning and end of the stance phase. Finally, the maximum and minimum values of the detrended data were summed to calculate height or speed changes (*Figure 4—figure supplement 1*).

## Nondimensional parameters

We chose to nondimensionalize a unit of mass by an animal's body mass (*m*), length by an animal's natural leg length (*R*), and acceleration by gravitational acceleration at the surface of the earth (*g*). Following this rule, we could nondimensionalize speed and spring constants as shown below:

$$Fr \equiv \frac{v^2}{Rg}$$

$$\gamma_s \equiv \frac{k_s R}{mg}$$

$$\gamma_a \equiv \frac{k_a}{mgR}$$

## System of ordinary differential equations for SLIP and ARSLIP and details regarding fitting ordinary differential equations to individual steps

The following system of ordinary differential equations (ODEs) (*Equations 6 and 7*) is derived using Euler–Lagrange equations to describe ARSLIP. A polar coordinate system was chosen for simplicity.

$$\ddot{r} = r\dot{\theta}^2 + \frac{k_S}{m}(R - r) - g\cos\theta \tag{6}$$

$$\ddot{\theta} = -\frac{2\dot{\theta}\dot{r}}{r} - \frac{k_a\theta}{mr^2} + \frac{g}{r}\sin\theta \tag{7}$$

where $r(t)$ is the length of the single effective leg used in SLIP and ARSLIP models, $\theta(t)$ is the leg angle from the vertical axis, $k_s$ is the leg spring constant, $k_a$ is an angular spring constant, $m$ is the total mass of a fly, and $R$ is the natural leg length. Dot denotes time derivative. A detailed derivation of the ODE is presented in the supplementary section of *Antoniak et al., 2019*.

The best fit of ARSLIP to a given experimental trajectory was found using the Global Search algorithm (*Dixon, 1978*) (MATLAB Global Optimization Toolbox). The RMSEs for height and distance were individually evaluated, and the sum was used as the objective function.

$k_s$ and $k_a$ were searched within the range below (*Equations 8 and 9*):

$$0 < k_s < 50\left(\frac{10^{-3}N}{m}\right) \tag{8}$$

$$0 < k_a < 50\left(10^{-9}N \cdot m\right) \tag{9}$$

These bounds were chosen empirically. $R$ and $m$ were experimentally measured for the various strains (*Table 3*). The length of the middle leg was measured from the still frames of the fly. Multiple measurements of a fly were averaged ($R_{real}$), and then based on the average value optimal $R$ was estimated by searching within ±10% boundary of $R_{real}$. Mass (m) was measured by averaging the mass of 10 individuals from the same genotype and gender.

Because a model with two effective legs would have too many parameters and would obscure many of the clear insights gained from modeling, we chose steps for which the duration of support by a single tripod was at least 25% of the tripod stance. This criterion does not mean that 75% of the step is spent with both tripods on the ground. Because the tripod legs are not synchronized, much of the time spent with both tripods on the substrate is the time it takes for legs from the second tripod to leave the ground. The experimentally measured initial conditions of $\theta$ and $\dot{r}$ were used to constrain the optimal initial condition. The optimal initial condition was constrained to be within ±10% of the measurements. Because we set the foothold location of ARSLIP as the middle of the front and hind leg foothold positions, initial conditions of $\theta$ and $r$ could be determined from experimental data.

The SLIP model was fitted using a similar method under the same parameter conditions except for the absence of $k_a$ due to the lack of angular spring in SLIP. The system of ODEs for SLIP is given by

$$\ddot{r} = r\dot{\theta}^2 + \frac{k_s}{m}(R - r) - g\cos\theta \tag{10}$$

**Table 3.** Measured parameter values for each species and gender.

| Fly number | Gender | Strain | $R_{real}$ (mm) | Mass (mg) |
|---|---|---|---|---|
| 1 | Female | Berlin K | 2.1032 | 1.0809 |
| 2 | Female | Berlin K | 2.184 | |
| 3 | Female | Berlin K | 2.1031 | |
| 4 | Female | Berlin K | 1.9988 | |
| 5 | Female | Berlin K | 1.9869 | |
| 6 | Male | Berlin K | 1.8888 | 0.6798 |
| 7 | Male | Berlin K | 1.9272 | |
| 8 | Male | Berlin K | 2.0646 | |
| 9 | Male | Berlin K | 2.0404 | |
| 10 | Female | W1118 | 2.042 | 1.123 |

$$\ddot{\theta} = -\frac{2\theta\dot{r}}{r} + \frac{g}{r}\sin\theta \tag{11}$$

For both models, the gravitational constant $g$ had a value of $9.807 m/s^2$.

## Derivation of the formula relating tripod model to ARSLIP

The total elastic potential energy of a springy tripod (*Figure 5*) is given by

$$V_{tri} = \frac{1}{2}k\left(R_{tri} - \sqrt{r^2 + L^2 - 2rL\sin\theta}\right)^2 + \frac{1}{2}k\left(R_{tri} - \sqrt{R_{tri}^2 + L^2 + 2rL\sin\theta}\right)^2 + \frac{1}{2}k(R_{tri} - r)^2 \tag{12}$$

The variables are also enumerated in *Table 1* and shown in *Figure 5*. In this equation, $R_{tri}$ is the natural length of the springy tripod that is optimized for each fly; $r$ is the height of the middle leg; $\theta$ is the angle that it makes with the vertical axis, which is identified with the radial and angular coordinate of ARSLIP; $L$ is the measured spread of the tripod (*Figure 7A*); and $k$ is the stiffness of a given leg. The total elastic potential energy is simply a sum of the potential energies due to each individual leg. We claim that $V_{tri}$ is equivalent to the ARSLIP potential energy, $V$ (*Equation 12*), for evolution that is close to the midpoint ($R_m$), and $\theta = 0$.

$$V = \frac{1}{2}k_s(R_{tri} - r)^2 + \frac{1}{2}k_a\theta^2 \tag{13}$$

Because $|r/r_m| \sim 0.1$ and $|\theta| \sim 0.2$, it should be sufficient to show that the two potential energies agree with each other up to the quadratic order in fluctuations around the fixed point. This means that the first and second derivatives with respect to $r$ and $\theta$ at $r = r_m$ and $\theta = 0$ are the same for both the potential energies. Specifically,

$$\frac{\partial V}{\partial r} = \frac{\partial V_{tri}}{\partial r} \tag{14}$$

$$\frac{\partial V}{\partial \theta} = \frac{\partial V_{tri}}{\partial \theta} \tag{15}$$

$$\frac{\partial^2 V}{\partial r^2} = \frac{\partial^2 V_{trip}}{\partial r^2} \tag{16}$$

$$\frac{\partial^2 V}{\partial \theta^2} = \frac{\partial^2 V_{trip}}{\partial \theta^2} \tag{17}$$

$$\frac{\partial^2 V}{\partial \theta \partial r} = \frac{\partial^2 V_{trip}}{\partial \theta \partial r} \tag{18}$$

are the same for both the potential energies. The relations involving the first derivative of $\theta$ (*Equation 15*) and cross-double derivative involving both $\theta$ and $r$ are automatically satisfied because of symmetry (*Equation 18*), the latter demonstrating the independence of the radial and angular springy forces that are assumed in the ARSLIP model.

We are then left with three equations, including the first derivative of $r$ (*Equation 14*), and the two double derivatives w.r.t. $r$ and $\theta$ (*Equations 16 and 17*). The effective ARSLIP potential energy involves the parameters $R$, $k_s$, and $k_a$. Consequently, it is possible to relate ARSLIP and springy tripod using the following equations:

$$k_s(R - r_m) = k\left[R_{tri} - R_m\left(3 - \frac{2R_{tri}}{\sqrt{L^2 + r_m^2}}\right)\right] \tag{19}$$

$$k_a = \frac{2kL^2 r_m^2 R_{tri}}{\left(L^2 + r_m^2\right)^{3/2}} \tag{20}$$

$$k_s = k\left[3 - \frac{2L^2 R_{tri}}{\left(L^2 + r_m^2\right)^{\frac{3}{2}}}\right] \tag{21}$$

Thus, the parameters, $R$, $k_s$, and $k_a$ can be determined from the tripod potential energy parameters, $k$ and $R_{tri}$, and the geometric quantities $r_m$, $L$ (*Equations 19–21*). We obtained $r_m$ and $L$ from the geometric data for each step. We assumed that a given fly has the same $R_{tri}$ and $k$, and fit $k_s$ and $k_a$ for all the steps of the given fly. This assumption led to a best fit value of $k$ and $R_{tri}$.

## Acknowledgements

We would like to acknowledge the members of Bhandawat lab and Sanjay Sane and his lab, and Michael Dickinson for critical comments on earlier versions of the manuscript. This research was supported by NIDCD (VB), NINDS (VB), and an NSF CAREER award (VB).

## Additional information

### Funding

| Funder | Grant reference number | Author |
| --- | --- | --- |
| National Science Foundation | IOS-1652647 | Vikas Bhandawat |
| National Institute on Deafness and Other Communication Disorders | RO1DC015827 | Vikas Bhandawat |
| National Institute of Neurological Disorders and Stroke | RO1NS097881 | Vikas Bhandawat |

The funders had no role in study design, data collection and interpretation, or the decision to submit the work for publication.

### Author contributions

Chanwoo Chun, Conceptualization, Data curation, Software, Formal analysis, Validation, Investigation, Visualization, Methodology, Writing - original draft, Writing - review and editing; Tirthabir Biswas, Conceptualization, Formal analysis, Validation, Writing - original draft, Writing - review and editing; Vikas Bhandawat, Conceptualization, Resources, Data curation, Supervision, Funding acquisition, Visualization, Methodology, Writing - original draft, Project administration, Writing - review and editing

## Author ORCIDs

Chanwoo Chun [ID] http://orcid.org/0000-0003-0759-6727
Vikas Bhandawat [ID] https://orcid.org/0000-0002-2608-0403

## Decision letter and Author response

Decision letter https://doi.org/10.7554/eLife.65878.sa1
Author response https://doi.org/10.7554/eLife.65878.sa2

# Additional files

## Data availability

Data are available on Dryad under doi:10.5061/dryad.m63xsj41g and Github https://github.com/vbhandawat/FlyTripod_eLife_2021/ copy archived at https://archive.softwareheritage.org/swh:1:rev:1dc429fc0da4cc5ff4f62617760447613f85980b/.

The following datasets were generated:

| Author(s) | Year | Dataset title | Dataset URL | Database and Identifier |
|---|---|---|---|---|
| Chun C, Biswas T, Bhandawat V | 2021 | Data from: *Drosophila* uses a tripod gait across all walking speeds, and the geometry of the tripod is important for speed control | http://dx.doi.org/10.5061/dryad.m63xsj41g | Dryad Digital Repository, 10.5061/dryad.m63xsj41g |
| Chun C, Biswas T, Bhandawat V | 2021 | FlyTripod_eLife_2021 | https://github.com/vbhandawat/FlyTripod_eLife_2021/ | Github, FlyTripod_eLife_2021 |

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

## Appendix

### What determines whether the gait will be fly-like or cockroach-like?

Whether the gait will be fly-like or cockroach-like is determined by the relative strengths of the radial and angular springs, which in turn depend on the tripod geometry and stiffness of the leg. In this Appendix, we will derive an expression that predicts fly-like versus cockroach-like gait based on the leg stiffness and tripod geometry.

From the ARSLIP model, we arrive at the following condition for negative horizontal acceleration at the midpoint, a characteristic of mid-stance velocity maxima:

$$\gamma_a > \gamma_s (1 - \bar{r}_m) \bar{r}_m, \tag{A1}$$

where $\bar{r}_m \equiv r_m/R$, $r_m$ is the mid-stance height, and $R$ is the natural length of the effective SLIP spring. This equation is derived from a consideration of forces at mid-stance. This is derived in the Antoniak et al. manuscript.

Assuming the empirical observation that while walking the mid-stance height of the fruit flies is very close to the leg lengths that balance gravity, we have

$$\gamma_s (1 - \bar{r}_m) = 1. \tag{A2}$$

Equation A2 simply means that the forces due to radial spring are close to the weight of the fly. *Equation A2*, in turn, implies that for the fly-like mid-stance maximum velocity profiles we must have

$$\gamma_a > \bar{r}_m \implies k_a > m g r_m \tag{A3}$$

To understand what this implies in terms of tripod leg characteristics and geometry, we need to rewrite the above inequality in terms of tripod quantities. Substituting $k_a$ from *Equation 20,* we have

$$\frac{2kL^2 r_m^2 R_{tri}}{\left(L^2 + r_m^2\right)^{\frac{3}{2}}} > m g r_m \implies \gamma > \frac{\left(\eta^2 + 1\right)^{\frac{3}{2}}}{2\eta^2} \tag{A4}$$

where $\eta \equiv \frac{L}{r_m}$ and $\gamma \equiv \frac{kR_{tri}}{mg}$.

Thus, the above inequality (*Equation A4*) provides a dividing curve in the $\eta - \gamma$ plane between velocity maxima (like flies) versus velocity minima (like cockroaches).

It is also illuminating to see how geometry shapes the angular and radial spring constants. From *Equations A2* and *21*, we have

$$\gamma_s = \gamma \left[ 1 + \frac{2}{\left(\eta^2 + 1\right)^{\frac{3}{2}}} \right] \tag{A5}$$

after some algebraic manipulations. From *Equation 20,* we then have

$$\gamma_a = \frac{2\gamma \bar{L}^{-2} \bar{r}_m^2}{\left(\bar{L}^{-2} + \bar{r}_{m,tri}^2\right)^{\frac{3}{2}}} \left(\frac{R_{tri}}{R}\right) = \frac{2\gamma \bar{L}^{-2} \bar{r}_m^2}{\left(\bar{L}^{-2} + \bar{r}_{m,tri}^2\right)^{\frac{3}{2}}} \left[ 3 - \frac{2\bar{L}^{-2}}{\left(\bar{L}^{-2} + \bar{r}_m^2\right)^{\frac{3}{2}}} \right] \frac{\gamma}{\gamma_s} \tag{A6}$$

Therefore, $\gamma_a$ depends on $\eta$, $\gamma$, $\bar{r}_{m,tri} \equiv \frac{r_m}{R_{tri}}$, and $\bar{L} \equiv \frac{L^2}{R_{tri}}$.

