## [Decision Letter]

**Acceptance summary:**

Understanding the biomechanics of movement across species - especially legged locomotion - has proven to be a challenging endeavor. This manuscript shows how relatively simple modifications to a classic model in the field can lead to an understanding of legged locomotion in fruit flies, whose center-of-mass dynamics during movement were previously unexplained, allowing for a re-evaluation of cross-species locomotion studies. In all, this is a carefully done study that will be of interest to researchers studying legged locomotion, and motor control more broadly.

**Decision letter after peer review:**

[Editors’ note: the authors submitted for reconsideration following the decision after peer review. What follows is the decision letter after the first round of review.]

Thank you for submitting your work entitled "*Drosophila* walks with a tripod gait, and the geometry of the tripod is important for the control of walking speed" for consideration by *eLife*. Your article has been reviewed by three peer reviewers, including Gordon Berman as the Reviewing Editor and Reviewer #1, and the evaluation has been overseen by a Senior Editor.

Our decision has been reached after consultation between the reviewers. Based on these discussions and the individual reviews below, we regret to inform you that your work will not be considered further for publication in *eLife*.

Although the reviewers felt that there was a great deal of merit in the ideas presented (particularly, using the ARSLIP model to understand the within-gait velocity profile), there was a general agreement that the manuscript was somewhat disjointed between the two halves - the gait distance metric and the ARSLIP model - and that the paper would be better suited for a more specialized journal, and perhaps even split into two different papers. Moreover, there were some systematic critiques of the gait distance metric implementation (in particular, see reviewer 2 and 3's comments), and the paper definitely needed to be put in the context of DeAngelis et al., 2019, which came to some similar conclusions with a different methodology.

Reviewer #1:

In this work, the authors use data from an automated tracking system to assess gait dynamics in *Drosophila*. In the first part of the paper, they claim that all of the locomotion dynamics can be explained via a modified tripod gait (introducing a Gait Distance metric), and in the second part, they introduce a new model, ARSLIP, for explaining the inverted speed profile seen in *Drosophila* locomotion (compared to cockroaches, for example).

In general, I thought that the second portion of the manuscript was model compelling than the first.

1) Specifically, the first part seems to cover similar intellectual terrain as DeAngelis et al., 2019, and so the authors need to place this work in the context of that paper's findings.

2) For the Gait Distance metric, I find it somewhat problematic that any noise will necessarily increase the value, but this is likely a minor concern.

3) In Figure 3C, although it is clear that any two pairs of speeds show no difference in L1-R3, it looks like there might be a trend going from low to high speeds. What does a linear regression show?

4) In Figure 3–S2, just because the real-time delays increase with cycle period doesn't by itself keep the phase delay constant – the delay should fall along a line with the same slope for all curves, no? If you do a best-fit slope to all four plots simultaneously (allowing for varying intercepts), what value do you get?

5) In general, I thought that the portion of the manuscript that explained the ARSLIP model was much stronger and clearer. A key scientific discrepancy was identified, and a plausible model was presented.

6) In Figure 6C, since it's predicted vs. actual, should the comparison not be to a linear fit, but instead to the line y = x?

Reviewer #2:

This manuscript from Chun and colleagues focuses on two key claims based on limb and body tracking in walking *Drosophila*. First, they analyze limb coordination patterns and propose that their data that supports a single, continuously modulated tripod gait being used over many walking speeds. Second, they propose a new biomechanical model for walking, which adds an angular restoring force to a widely used inverted spring model for walking. This new model explains a key aspect of their center of mass tracking data, in which forward velocity peaks as height also peaks, which cannot be explained by previous models.

I had a few concerns about this work. The first is related to the Materials and methods, where the authors describe their distance metric. My reading is that this branch cut choice means that phase a difference of +0.45 cycles and one of +0.55 cycles (goes to -0.45) would look further apart than +.45 and 0 cycles (distance = 0.9 vs. 0.45). This is a pretty worrisome property of this analysis metric, and I would think that it could cause serious issues. I understand that this was chosen for computing distances expected to be 0 for tripod, but wouldn't it create some large, unintended errors in some plausible non-tripod cases? Much of the paper is based on this distance metric, so it's critical that it have properties that don't lead to interpretation errors. I don't even see why one needs a branch cut – isn't it possible to use a trigonometric function to compute distances to eliminate this issue? For instance, cos(\δ \phi) goes from 1 to -1, depending on how far out of phase limbs are, but has no branch cut issues.

Secondly, I'm not sure that the claims in the first portion of the paper need to be so strong in order to support the second half of the paper. For instance, and the authors say as much, wouldn't a putative tetrapod gait support much the same model? I just don't know what to say about example traces, like those in Mendes et al. Figure 4B, which look pretty different from tripod, over time, for many steps. It seems clearly not tripod. Perhaps it's not typical, but it seems different from just sloppy, since the phase differences among limbs are retained over many steps. It does not appear to me to be tripod with longer stances, as this paper would seem to argue.

Last, some of the ground in the first portion of this paper has been covered by a study on similar topics that was unreferenced here (DeAngelis et al., 2019). Like this study, that paper looked at gaits using phase relationships rather than swing/stance counts and found (from my reading) some similar results. However, the 2019 paper had a quite different explanation than "always tripod". How do these results fit together? The 2019 paper suggests a model for generating gaits that can produce tripod-like, wave-like, and tetrapod-like patterns, but is not just a sloppy tripod at slow walking speeds. How does that claim relate to the claims made here?

This paper also did not reference Szczecinski et al., 2018, which showed how a continuum of walking coordination patterns in *Drosophila* might benefit stability. Given that the second portion of this manuscript is biomechanical and the restoring force is related to stability, it would be important to mention this and discuss how these results relate to those ones.

Reviewer #3:

In this work, the authors introduce a metric based on phase deviation from a set of “template” gaits to robustly classify the walking gaits. Many classical, as well as more recent, works in the literature have employed gait diagrams, which indicate the number of legs in contact with the ground, as the metric to characterise and classify locomotor behaviors. It has been a timely contribution to point out that these metrics fall short in robustly classifying gaits and being sensitive to dynamical variations in leg coordination patterns. However, I have concerns regarding the authors' incomplete assessment of the literature.

1) One crucial reference is the recent e*Life* paper by DeAngelis et al. (2019) which presents a rather exhaustive analysis of *Drosophila* walking. Importantly, DeAngelis et al. classify gait based on the distance of leg phases from a “template phase”, as quantified by a modified Kuramoto coherence index, see Equation 5 in their Materials and methods. This is very similar to the proposed “gait distance”, the only difference being that Equation 5 in DeAngelis et al. quantifies phase distance on the unit sphere, whereas the “gait distance” uses the Euclidean distance. Thus, the authors should demonstrate how the metric of DeAngelis performs in comparison to the “gait distance” on the same dataset. This would either justify or call into question the utility of the “gait distance” metric rather than the metric used by DeAngelis et al.

2) Another critique is that it seems in the “gait distance” one should a priori know the phases of which legs are to be compared against which legs of the template. This could make the metric difficult to apply in an unbiased way and may introduce a variation in the results as well as change the sensitivity of the metric when the phases of additional (perhaps all) leg pairs are compared simultaneously. The second claim of the authors' is that the only coordinated walking pattern that *Drosophila* adopts at a wide range of walking speeds is consistent with the tripod gait and slight phase-perturbations thereof. This raises another concern. Although DeAngelis et al. also concludes the abundance of the tripod gait (based on their independent dataset), their metric also detects other canonical gaits as the speed is varied. Such sensitivity is a desirable feature of any useful metric. Thus, the fact that the “gait distance” does not detect other gaits can also be explained by its lack of sensitivity in addition to those gaits not being present in the data. For it to be a useful metric, the authors should demonstrate that “gait distance” generalizes to other gaits and detect gait transitions.

3) In my view, the main contribution of the paper is the parsimonious kinetic model ARSLIP to describe the tripod gait. A large part of the *Drosophila* literature assumes that (not just) walking behaviors follow feedforward, central pattern generator driven dynamics, giving little importance to mechanical inputs. That the motion of the CoM can be described by a purely reaction-based, lossless spring model is a useful advance. However, I would like to see the authors perform a more careful analysis of the ARSLIP model and consider its predictions. For example, the model hinges the fact that the dynamics of the springy tripod (Figure 7) can be reduced to an effective angular-radial spring pair. However, the springy tripod can have multiple dynamical modes of vibration, including lateral and longitudinal components, and moreover, due to its lateral asymmetry these modes can have strong lateral components. It is unclear to me whether the authors study these steady state vibration modes and how they are consistent with the CoM motion during stable locomotor patterns, or just consider the motion of the suspended mass in the initial transient phase. Thus, the authors should consider predictions in the X and Y planes (not just Z). This assumption could be justified through simple simulations. Second, the ARSLIP model is only valid at small perturbations from the stable state. However, the authors say that CoM motion during locomotion yields a 50% compression of the spring, which is no longer in the regime of validity. Third, although the authors claim that this simplified model describes fly locomotion, they only demonstrate it on the motion of the CoM. One prediction the authors could make is taking the model fitted to the CoM motion and using it to predict resulting changes in leg coxa-to-pretarsus distance, which could be verified from their experimental data. Fourth, it would be useful to describe how the model could be generalized to generate multiple gaits.

4) Finally, the paper could be improved dramatically by carefully defining model parameters where they are used and describing the model fitting procedure with more clarity. This would ensure reproducibility for future studies by the community.

[Editors’ note: further revisions were suggested prior to acceptance, as described below.]

Thank you for submitting your work entitled "*Drosophila* walks with a tripod gait across all speeds, and the geometry of the tripod is important for speed control" for consideration by *eLife*. Your article has been reviewed by three peer reviewers, including Gordon J Berman as the Reviewing Editor and Reviewer #1, and the evaluation has been overseen by a Senior Editor.

Our decision has been reached after consultation between the reviewers. Based on these discussions and the individual reviews below, we regret to inform you that your work will not be considered further for publication in *eLife*.

While the reviewers all thought there was strength in the second half of the paper involving the ARSLIP model, they also all agreed that the first portion of the paper involving the gait distance metric was better suited for a more specialized journal. In particular, the detailed technical analyses and arguments around gait definition appear not reader-oriented and a bit defensive, more an argument made with a crowd of experts rather than to a general scientific readership.

We suggest that the authors submit a new paper containing a very brief section on kinematic analysis (just showing that the tripod leg coordination is largely conserved and helping to motivate and set up the potential generality of the ARSLIP model), and an expanded modeling analysis that includes additional comparisons to the previous literature to better show the potential generality of the model. A paper of this nature would make a significant contribution and could potentially be accepted at *eLife*.

Reviewer #1:

I think that this submission improves many aspects over the previous iteration, although I would have like to have seen the focus much more on the AR-SLIP model, which, in my opinion, is the key contribution of this article. Accordingly, I will focus my comments on the Introduction and the Gait Distance portion of the article, which I still believe needs some re-framing.

1) The Introduction is far too long and it is difficult to isolate the key findings/context of the paper. I have now read versions of this manuscript several times, and I still had difficultly parsing through all of the text. While I think that many of the aspects here would make parts of an excellent review article, I encourage the authors to consider shortening the Introduction by a factor of two or more.

2) Although I think that the authors are correct in their critique of trying to use the DeAngelis gait metric for tetrapod and metachronal gaits, I think that it's somewhat missing the point. The point of DeAngelis et al. is not that all gaits are tripod, but that there is a continuous manifold underlying gait coordination in forward walking. The authors arrive at a similar conclusion here, but through different means. To be clear, given that the two articles were initially submitted around the same time, I don't consider this a "primacy" issue, but I think that it would be far more effective and useful to the community to discuss the result in this light.

3) More broadly, I think that the argument about tripod/tetrapod/metachronal definitions to be a bit too specific for an *eLife* audience. I recognize that this is the language that has been used in past articles (including *eLife* articles), but these are just words, and the authors correctly point out here that the words can alter their meaning depending on their precise mathematical definition. Instead of focusing on this field-focused argument (which would be more suited for a more biomechanics-focused journal), I ask the authors to focus more on the geometry of the space of movements, rather than on the nomenclature.

4) A more technical question I had is that I was not sure why only four phase differences are used in the gait distance measure. Why not use all 21 potential comparisons?

5) In Figure 6F (caption), do you mean that b = -0.55? Also, when fitting a power law, this should always be done on a log-log plot, and given that there is only one order of magnitude in each direction, it is very difficult to distinguish between even an exponential and a power law with these data (let alone measure an exponent). Also, why is there no constant term (y = ax^b^ +c)? It is not clear to me that the stance time necessarily needs to go to zero at very fast speeds.

6) Similarly, in Figures 6C-D, is there any evidence that a quadratic/power law fit is more suitable than a linear fit? Some type of model selection criterion should be used.

Reviewer #2:

We would like to thank the authors for making an effort to improve their manuscript. The content is now better motivated, and the contribution is sharpened. Although we appreciate the increased quality of the writing as well as the better coherence between the first and second sections, the readability could be substantially improved by condensing the writing and eliminating repetition. In particular, the Introduction is very long, and should be focused.

We still feel that the main contribution of the paper is in the second section describing the ARSLIP model. Parsimonious models are extremely valuable for a qualitative understanding of dynamics. The ARSLIP model is novel and provides useful thought experiments, some of which the authors touch upon in the Discussion.

The remainder of this review refers to the first section, analyzing gaits.

Regarding DeAngelis et al. we can corroborate that there is a sign error in their coherence metric; the plus (+) sign should be a minus (-) sign, i.e., the formula should measure phase differences between template and measures phases. So, we agree it would be difficult to make any objective comparison. However, we still have two major concerns.

First, we noticed that the coherence metric of DeAngelis, with the sign corrected, is exactly the same at the "gait angle" that the authors now propose as their own (i.e., without citation) in the revised manuscript (Equation 10 here vs Equation 5 in DeAngelis). There does not seem to be any basis for introducing this metric, or any reference to its original description-however erroneous. We therefore urge the authors to omit this section altogether.

Second, we previously raised the concern that "one should a priori know the phases of which legs are to be compared to which legs of the template." This has not been addressed. For the authors' metric to be useful for a broad range of animals and gaits, one would ideally compare as many leg pairs as possible (or all) since not accounting for phase differences between some relevant leg pairs may fail to distinguish some gaits. But doing so can decrease the sensitivity of their gait distance metric. This is because gait distance (Equation 5) measures the straight-line distance between phases (by Euclidean metric) and not the geodesic (manifold) distance along the natural manifold on the unit sphere (like the DeAngelis metric or the gait angle, Equation 12). Another way to see this is that Taylor expanding Equation 12 leads to Equation 5 – thus the gait distance (5) is a special case for small phase differences of (12), which is when straight-line distance is approximately equal the manifold distance. We therefore expect that increasing the dimensionality (including more leg pairs) would reduce sensitivity of the metric for large phase differences. In their response, the authors presented a comparison between two metrics; the gait distance (5) with specific leg pairs and the gait angle (10) with all leg pairs. But this is not relevant because two different metrics are being compared. We would like to ask the authors to explore the above potential limitation by simply including progressively more leg pairs in the computation of the gait distance – same metric – and focusing on the statistical significance between distinguishing tetrapod and a tripod gaits where large phase differences are to be expected.

Reviewer #3:

This study carefully investigated the gait of fruit fly terrestrial locomotion, focusing on how stance and swing patterns of the six legs change as locomotion speed varies. It was found that the animal predominantly uses a tripod gait at all but the lowest speeds. The authors proposed a gait distance metric to quantify the gait. In addition, they developed a novel angular and radial SLIP model (ARSLIP) as a simple template to explain the mechanics of the tripod gait. A surprising insight from the simple model is that flies can simply change the geometry of its leg tripod to achieve control of effective leg stiffness, without having to stiffen up each leg.

Overall, this is a comprehensive study on the subject and the manuscript is well written. I am most excited about the ARSLIP model, which, with further testing and validation in other species with rigorous experiments and quantitative data, can potentially provide a general model to explain terrestrial locomotion of a diversity of insects. I am supportive that the paper should be further considered for publication in *eLife*, but I do have one major comment that the authors should address or argue against.

The authors should make it very clear up front what they mean by "gait". It appears that, when the authors stated that fruit fly uses pretty much the same tripod gait across speeds, they are really just stating that the fruit fly uses basically the same leg coordination to move (which even the authors acknowledged) regardless of what speed they move at and how their legs come in contact with the ground. I think this is true and well supported by their data. However, I am not sure if it is really that important to stress and try to convince everyone that this is THE best/right way to define gait. Frankly, I think the authors should simply state that they found leg "coordination" to be relatively constant, not "gait", which can be defined/interpreted differently by different people or for different purposes.

It is well known that there are different ways of defining gait for terrestrial locomotion. For example, a commonplace definition is by the sequence/phase and duty factor that the feet are in contact with the ground. Alternatively, center of mass (CoM) dynamics can be used to define gait instead. With these different ways, what appear as walking (e.g., cockroach using an alternating tripod footfall pattern with a >0.5 duty factor) using the first definition may in fact be classified as SLIP-like running when CoM dynamics are evaluated.

The authors have carefully compared their definition of gait with that in a few closely-related recent studies. The authors' gait definition is mainly based on kinematics (coordination to be exact) of the legs themselves, whereas the other studies used stance and swing phase patterns (which are footfall patterns on the ground). These are different ways of defining gait from kinematics. While I agree with the technical aspects of the comparisons, as well as some flaws that the authors have identified in the other studies, I am not sure if leg coordination alone is THE best/right way of defining gait.

As an example, we can again consider the cockroach case above. Using the author's definition, cockroaches also only demonstrates a single tripod "gait" in terms of leg coordination when speed increase from very small to very large. Does this mean that all the work defining cockroach walking at low speeds and running at high speeds using CoM dynamics are wrong in having such definitions?

In my opinion, the authors have shown convincing evidence that leg coordination (which they refer to as "gait") does not change significantly as speed increases. The resulting kinematics of stance and swing phases do vary with speed, and other researchers (or the authors hypothetically) may choose to use those instead to define gait. Frankly, I do not think this is the key advancement of the study and am not sure it is worth this much effort to try to convince people of different opinions (although I certainly understand that the authors may be choosing to emphasize this in order to address the reviewers' comments).

I think the authors should highlight the contribution they are making with the ARSLIP model. Again, as I said above, this can potentially be a general model and a major advancement of the field of terrestrial animal locomotion. I see this model (with further validation and generalization beyond this study) as the equivalent of seminal work on bipedal compliant leg template model, which unifies SLIP running and inverted pendulum walking for bipedal locomotion (Geyer, Seyfarth and Blickhan, 2006). The authors cited this work, but do not seem to appreciate its importance in the area of bipedal locomotion, or how their own work has the potential of being the same. The authors' model provides initial insight why hexapedal animals can use the same set of legs to achieve dynamic locomotion over a wide range of speeds.

The authors do provide evidence that the velocity profile of CoM dynamics are opposite to SLIP-like, which is another thing that motivates the ARSLIP model. I think this should be also emphasized more to set up the model.

I think the authors are missing an opportunity not to discuss this more, and their paper will improve substantially by building around this central idea, which the kinematic data and analysis provide compelling evidence for. I think this will significantly increase the impact of the paper beyond the field of animal locomotion. It is these kinds of simple, general biomechanical/dynamic models that have provided the foundation for simple yet robust robots such as RHex, Atlas, BigDog, etc. Personally, I think that this is something that many people will remember the study for, not so much the technical debate on which way of defining "gait" from pure kinematics is better, which are concerns more for the specialists.

Therefore, I strongly encourage the authors to cut down on the debate of "gait" and highlight the modeling contributions and elaborate (at least speculate) what should be done in future to test and validate it as a general model.

---

## [Author Response]

[Editors’ note: the authors resubmitted a revised version of the paper for consideration. What follows is the authors’ response to the first round of review.]

Reviewer #1:In this work, the authors use data from an automated tracking system to assess gait dynamics in *Drosophila*. In the first part of the paper, they claim that all of the locomotion dynamics can be explained via a modified tripod gait (introducing a Gait Distance metric), and in the second part, they introduce a new model, ARSLIP, for explaining the inverted speed profile seen in *Drosophila* locomotion (compared to cockroaches, for example).In general, I thought that the second portion of the manuscript was model compelling than the first.1) Specifically, the first part seems to cover similar intellectual terrain as DeAngelis et al., 2019, and so the authors need to place this work in the context of that paper's findings.

As noted earlier, our manuscript has been in the public domain for more than a year, and we think of our work as contemporaneous to the DeAngelis work. Moreover the DeAngelis work has basic issues that make it difficult for us to compare their study to ours.

2) For the Gait Distance metric, I find it somewhat problematic that any noise will necessarily increase the value, but this is likely a minor concern.

We can understand the reviewer’s viewpoint. But we believe the increase is a useful feature because it quantifies how close to the ideal gait, a metric is.

3) In Figure 3C, although it is clear that any two pairs of speeds show no difference in L1-R3, it looks like there might be a trend going from low to high speeds. What does a linear regression show?

Yes. This is an excellent point. We have changed our analysis. Instead of making comparisons between neighboring bins, we are performing linear regression. The linear regression shows a small but significant change in phase.

4) In Figure 3–S2, just because the real-time delays increase with cycle period doesn't by itself keep the phase delay constant – the delay should fall along a line with the same slope for all curves, no? If you do a best-fit slope to all four plots simultaneously (allowing for varying intercepts), what value do you get?

Yes. This is another excellent point. We now show that the time difference between legs changes in a manner consistent with phase changes.

5) In general, I thought that the portion of the manuscript that explained the ARSLIP model was much stronger and clearer. A key scientific discrepancy was identified, and a plausible model was presented.

Thanks! We have worked hard to connect the two parts of the manuscript.

6) In Figure 6C, since it's predicted vs. actual, should the comparison not be to a linear fit, but instead to the line y = x?

We think that the reviewer is referring to Figure 9C. If so, yes, the line is not a fit but the line of unity or x=y.

Reviewer #2:This manuscript from Chun and colleagues focuses on two key claims based on limb and body tracking in walking *Drosophila*. First, they analyze limb coordination patterns and propose that their data that supports a single, continuously modulated tripod gait being used over many walking speeds. Second, they propose a new biomechanical model for walking, which adds an angular restoring force to a widely used inverted spring model for walking. This new model explains a key aspect of their center of mass tracking data, in which forward velocity peaks as height also peaks, which cannot be explained by previous models.I had a few concerns about this work. The first is related to the Materials and methods, where the authors describe their distance metric. My reading is that this branch cut choice means that phase a difference of +0.45 cycles and one of +0.55 cycles (goes to -0.45) would look further apart than +.45 and 0 cycles (distance = 0.9 vs. 0.45). This is a pretty worrisome property of this analysis metric, and I would think that it could cause serious issues. I understand that this was chosen for computing distances expected to be 0 for tripod, but wouldn't it create some large, unintended errors in some plausible non-tripod cases? Much of the paper is based on this distance metric, so it's critical that it have properties that don't lead to interpretation errors. I don't even see why one needs a branch cut – isn't it possible to use a trigonometric function to compute distances to eliminate this issue? For instance, cos(\δ \phi) goes from 1 to -1, depending on how far out of phase limbs are, but has no branch cut issues.

The reviewer is correct regarding the problem with the branchpoint. But, the branchpoint issue only relates to gait distance measures using stance start times. There are no such issues with the gait distance metric using phases. In response to the reviewer’s comment, we have now used the gait distance metric using phases as the primary measure of gait.

Secondly, I'm not sure that the claims in the first portion of the paper need to be so strong in order to support the second half of the paper. For instance, and the authors say as much, wouldn't a putative tetrapod gait support much the same model? I just don't know what to say about example traces, like those in Mendes et al. Figure 4B, which look pretty different from tripod, over time, for many steps. It seems clearly not tripod. Perhaps it's not typical, but it seems different from just sloppy, since the phase differences among limbs are retained over many steps. It does not appear to me to be tripod with longer stances, as this paper would seem to argue.

This is an excellent point. The reviewer is correct that we don’t need the gait to be tripod to perform the analysis in the second part of the manuscript. More importantly, the figure that the reviewer refers to convinced us to take another look at our gait analysis. Thank you! We do find examples of tetrapods in our data. Our overall conclusion remains the same – that tripod is the predominant gait, but our gait analysis is more complete with the inclusion of the tripod.

This paper also did not reference Szczecinski et al., 2018, which showed how a continuum of walking coordination patterns in *Drosophila* might benefit stability. Given that the second portion of this manuscript is biomechanical and the restoring force is related to stability, it would be important to mention this and discuss how these results relate to those ones.

We did discuss the Szczecinski study. This discussion has been further clarified. But we cannot go further into the question of stability because it is somewhat tangential to this study. The issue of stability is important to consider in future work.

Reviewer #3:In this work, the authors introduce a metric based on phase deviation from a set of “template” gaits to robustly classify the walking gaits. Many classical, as well as more recent, works in the literature have employed gait diagrams, which indicate the number of legs in contact with the ground, as the metric to characterise and classify locomotor behaviors. It has been a timely contribution to point out that these metrics fall short in robustly classifying gaits and being sensitive to dynamical variations in leg coordination patterns. However, I have concerns regarding the authors' incomplete assessment of the literature.1) One crucial reference is the recent eLife paper by DeAngelis et al. (2019) which presents a rather exhaustive analysis of *Drosophila* walking. Importantly, DeAngelis et al. classify gait based on the distance of leg phases from a “template phase”, as quantified by a modified Kuramoto coherence index, see Equation 5 in their Materials and methods. This is very similar to the proposed “gait distance”, the only difference being that Equation 5 in DeAngelis et al. quantifies phase distance on the unit sphere, whereas the “gait distance” uses the Euclidean distance. Thus, the authors should demonstrate how the metric of DeAngelis performs in comparison to the “gait distance” on the same dataset. This would either justify or call into question the utility of the “gait distance” metric rather than the metric used by DeAngelis et al.

Our manuscript has been in the public domain for more than a year and was first submitted to *eLife* in March 2019 – well before the DeAngelis manuscript. Moreover, the two metrics produce the same results. This similarity is further demonstrated in our manuscript.

2) Another critique is that it seems in the “gait distance” one should a priori know the phases of which legs are to be compared against which legs of the template. This could make the metric difficult to apply in an unbiased way and may introduce a variation in the results as well as change the sensitivity of the metric when the phases of additional (perhaps all) leg pairs are compared simultaneously. The second claim of the authors' is that the only coordinated walking pattern that *Drosophila* adopts at a wide range of walking speeds is consistent with the tripod gait and slight phase-perturbations thereof. This raises another concern. Although DeAngelis et al. also concludes the abundance of the tripod gait (based on their independent dataset), their metric also detects other canonical gaits as the speed is varied. Such sensitivity is a desirable feature of any useful metric. Thus, the fact that the “gait distance” does not detect other gaits can also be explained by its lack of sensitivity in addition to those gaits not being present in the data. For it to be a useful metric, the authors should demonstrate that “gait distance” generalizes to other gaits and detect gait transitions.

There are three points here: 1. Choosing legs is a feature, not a bug. It makes sense to choose legs whose phases are expected to change with a given gait transition. Choosing the legs does not predetermine the outcome in any way. In any case, one does not have to choose particular legs when calculating gait distance; all five phase differences can be employed.

We have also shown that whether we employ gait distance with a subset of legs (Figure 5), or gait angle with all the legs, the result is very similar.

We have discussed the results in the DeAngelis manuscript vs. our manuscript at length at the beginning of this critique. Briefly, there is no “sensitivity” issue with our gait metric. Our gait metric detects a tetrapod just fine. It is just that 96% of the steps are tripod. We have demonstrated this point in the current version of the manuscript by explicitly investigating steps that are not tripod.

3) In my view, the main contribution of the paper is the parsimonious kinetic model ARSLIP to describe the tripod gait. A large part of the *Drosophila* literature assumes that (not just) walking behaviors follow feedforward, central pattern generator driven dynamics, giving little importance to mechanical inputs. That the motion of the CoM can be described by a purely reaction-based, lossless spring model is a useful advance. However, I would like to see the authors perform a more careful analysis of the ARSLIP model and consider its predictions. For example, the model hinges the fact that the dynamics of the springy tripod (Figure 7) can be reduced to an effective angular-radial spring pair. However, the springy tripod can have multiple dynamical modes of vibration, including lateral and longitudinal components, and moreover, due to its lateral asymmetry these modes can have strong lateral components. It is unclear to me whether the authors study these steady state vibration modes and how they are consistent with the CoM motion during stable locomotor patterns, or just consider the motion of the suspended mass in the initial transient phase. Thus, the authors should consider predictions in the X and Y planes (not just Z). This assumption could be justified through simple simulations. Second, the ARSLIP model is only valid at small perturbations from the stable state. However, the authors say that CoM motion during locomotion yields a 50% compression of the spring, which is no longer in the regime of validity. Third, although the authors claim that this simplified model describes fly locomotion, they only demonstrate it on the motion of the CoM. One prediction the authors could make is taking the model fitted to the CoM motion and using it to predict resulting changes in leg coxa-to-pretarsus distance, which could be verified from their experimental data. Fourth, it would be useful to describe how the model could be generalized to generate multiple gaits.

The reviewer makes several important points here. We agree with some of these points and do not agree with others. One broader point is that a detailed mechanical analysis of every aspect of the model is beyond the scope of this study. We think that this model is a useful one and will continue to inform the links between neuroscience and mechanics and hopefully will inform many other studies. Regarding the specific issues raised:

1) Various modes of the springy tripod: There might be some utility in this analysis, but they must be data-driven. We have not only shown that ARLSIP is an accurate approximation, but also demonstrated that the shape of the tripod influences the mechanical stiffness of the system. We have also demonstrated (See Section 3.11 of the Materials and methods) that the radial and angular components are independent. The same holds for movement in the horizontal plane. This independence has already been demonstrated in the LLS model employed in the cockroach (J. Schmitt and P. Holmes. Mechanical models for insect locomotion: dynamics and stability in the horizontal plane I. Theory. *Biol Cybern*, 83:501–515, 2000.). This independence is likely to hold even more in flies because the lateral movement is relatively small.

2) Small perturbation limits: The 50% number comes from how much compression occurs through the weight of the fly. During walking, the ARSLIP radial spring is not compressing to its natural length. Rather it is oscillating a small amount <10% about its fixed point (*R*- mg/k_s_). So, the small perturbation limits very much hold.

3) Leg to pre-tarsus distance: This is an interesting idea, but for this analysis we will have to track the coxa-body joint, which is impossible to do with our current dataset because that joint is not visible on both camera views. However, we would like to explore this in future work.

4) Multiple gaits: Yes, we have done so in the last section of the Discussion. This investigation is going to be an important future direction. We will be exploring this idea in the future.

4) Finally, the paper could be improved dramatically by carefully defining model parameters where they are used and describing the model fitting procedure with more clarity. This would ensure reproducibility for future studies by the community.

Thank you for this suggestion. We have included a table with model parameters and beefed up our discussion of the model.

[Editors’ note: what follows is the authors’ response to the second round of review.]

Reviewer #1:I think that this submission improves many aspects over the previous iteration, although I would have like to have seen the focus much more on the AR-SLIP model, which, in my opinion, is the key contribution of this article. Accordingly, I will focus my comments on the Introduction and the Gait Distance portion of the article, which I still believe needs some re-framing.

We thank the reviewer for his careful reading of the manuscript. We have now focused primarily on the ARSLIP model.

1) The Introduction is far too long and it is difficult to isolate the key findings/context of the paper. I have now read versions of this manuscript several times, and I still had difficultly parsing through all of the text. While I think that many of the aspects here would make parts of an excellent review article, I encourage the authors to consider shortening the Introduction by a factor of two or more.

The Introduction has been reformatted and reduced by a factor of two.

2) Although I think that the authors are correct in their critique of trying to use the DeAngelis gait metric for tetrapod and metachronal gaits, I think that it's somewhat missing the point. The point of DeAngelis et al. is not that all gaits are tripod, but that there is a continuous manifold underlying gait coordination in forward walking. The authors arrive at a similar conclusion here, but through different means. To be clear, given that the two articles were initially submitted around the same time, I don't consider this a "primacy" issue, but I think that it would be far more effective and useful to the community to discuss the result in this light.

All the gait analysis has been deleted in the reformatting of the manuscript.

3) More broadly, I think that the argument about tripod/tetrapod/metachronal definitions to be a bit too specific for an eLife audience. I recognize that this is the language that has been used in past articles (including eLife articles), but these are just words, and the authors correctly point out here that the words can alter their meaning depending on their precise mathematical definition. Instead of focusing on this field-focused argument (which would be more suited for a more biomechanics-focused journal), I ask the authors to focus more on the geometry of the space of movements, rather than on the nomenclature.

All the gait analysis material has been removed. We will keep the reviewer’s critique in mind when publishing a detailed gait analysis in a separate manuscript.

4) A more technical question I had is that I was not sure why only four phase differences are used in the gait distance measure. Why not use all 21 potential comparisons?

All the gait analysis stuff has been removed. We have done this analysis and shown it in one of previous version of the manuscript. Once again, we might revisit this idea in the future.

5) In Figure 6F (caption), do you mean that b = -0.55? Also, when fitting a power law, this should always be done on a log-log plot, and given that there is only one order of magnitude in each direction, it is very difficult to distinguish between even an exponential and a power law with these data (let alone measure an exponent). Also, why is there no constant term (y = ax^b^ +c)? It is not clear to me that the stance time necessarily needs to go to zero at very fast speeds.

Thanks for this suggestion. We have indeed fit the data in Figure 6F (Now Figure 4F) using the expression suggested by the reviewer above.

6) Similarly, in Figures 6C-D, is there any evidence that a quadratic/power law fit is more suitable than a linear fit? Some type of model selection criterion should be used.

Thanks for this suggestion. Yes, there is no need for a quadratic fit in Figure 6C (Now Figure 4C), we have replaced this with a linear fit. A power law form is more appropriate for Figure 6D (now Figure 4D), and we have shown that form is warranted.

Reviewer #2:We would like to thank the authors for making an effort to improve their manuscript. The content is now better motivated, and the contribution is sharpened. Although we appreciate the increased quality of the writing as well as the better coherence between the first and second sections, the readability could be substantially improved by condensing the writing and eliminating repetition. In particular, the Introduction is very long, and should be focused.

We thank the reviewer for his careful reading. We have completely reformatted the Introduction making it almost half the number of words as before.

We still feel that the main contribution of the paper is in the second section describing the ARSLIP model. Parsimonious models are extremely valuable for a qualitative understanding of dynamics. The ARSLIP model is novel and provides useful thought experiments, some of which the authors touch upon in the Discussion.

We have taken the reviewer’s advice and limited our description of our work on gait.

The remainder of this review refers to the first section, analyzing gaits.Regarding DeAngelis et al. we can corroborate that there is a sign error in their coherence metric; the plus (+) sign should be a minus (-) sign, i.e., the formula should measure phase differences between template and measures phases. So, we agree it would be difficult to make any objective comparison. However, we still have two major concerns.

Thanks for corroborating the sign error.

First, we noticed that the coherence metric of DeAngelis, with the sign corrected, is exactly the same at the "gait angle" that the authors now propose as their own (i.e., without citation) in the revised manuscript (Equation 10 here vs Equation 5 in DeAngelis). There does not seem to be any basis for introducing this metric, or any reference to its original description-however erroneous. We therefore urge the authors to omit this section altogether.

Thanks for this comment. It is useful to us for the future. Dealing with the DeAngelis gait metric was difficult for us. It was not clear whether the sign error was a mistake or whether they were trying something different. Moreover, we did not think that we needed to explain this because distance and angle are obvious choices when comparing high-dimensional vectors. In neural coding, one does analysis both ways just to show robustness of the conclusion. But we will keep the reviewer’s comment in mind when we discuss this issue in the future.

Second, we previously raised the concern that "one should a priori know the phases of which legs are to be compared to which legs of the template." This has not been addressed. For the authors' metric to be useful for a broad range of animals and gaits, one would ideally compare as many leg pairs as possible (or all) since not accounting for phase differences between some relevant leg pairs may fail to distinguish some gaits. But doing so can decrease the sensitivity of their gait distance metric. This is because gait distance (Equation 5) measures the straight-line distance between phases (by Euclidean metric) and not the geodesic (manifold) distance along the natural manifold on the unit sphere (like the DeAngelis metric or the gait angle, Equation 12). Another way to see this is that Taylor expanding Equation 12 leads to Equation 5 – thus the gait distance (5) is a special case for small phase differences of (12), which is when straight-line distance is approximately equal the manifold distance. We therefore expect that increasing the dimensionality (including more leg pairs) would reduce sensitivity of the metric for large phase differences. In their response, the authors presented a comparison between two metrics; the gait distance (5) with specific leg pairs and the gait angle (10) with all leg pairs. But this is not relevant because two different metrics are being compared. We would like to ask the authors to explore the above potential limitation by simply including progressively more leg pairs in the computation of the gait distance – same metric – and focusing on the statistical significance between distinguishing tetrapod and a tripod gaits where large phase differences are to be expected.

This is an excellent point that we will definitely explore this point in the future.

Reviewer #3:This study carefully investigated the gait of fruit fly terrestrial locomotion, focusing on how stance and swing patterns of the six legs change as locomotion speed varies. It was found that the animal predominantly uses a tripod gait at all but the lowest speeds. The authors proposed a gait distance metric to quantify the gait. In addition, they developed a novel angular and radial SLIP model (ARSLIP) as a simple template to explain the mechanics of the tripod gait. A surprising insight from the simple model is that flies can simply change the geometry of its leg tripod to achieve control of effective leg stiffness, without having to stiffen up each leg.Overall, this is a comprehensive study on the subject and the manuscript is well written. I am most excited about the ARSLIP model, which, with further testing and validation in other species with rigorous experiments and quantitative data, can potentially provide a general model to explain terrestrial locomotion of a diversity of insects. I am supportive that the paper should be further considered for publication in eLife, but I do have one major comment that the authors should address or argue against.

We thank the reviewer for recognizing the potential importance of ARSLIP model.

The authors should make it very clear up front what they mean by "gait". It appears that, when the authors stated that fruit fly uses pretty much the same tripod gait across speeds, they are really just stating that the fruit fly uses basically the same leg coordination to move (which even the authors acknowledged) regardless of what speed they move at and how their legs come in contact with the ground. I think this is true and well supported by their data. However, I am not sure if it is really that important to stress and try to convince everyone that this is THE best/right way to define gait. Frankly, I think the authors should simply state that they found leg "coordination" to be relatively constant, not "gait", which can be defined/interpreted differently by different people or for different purposes.It is well known that there are different ways of defining gait for terrestrial locomotion. For example, a commonplace definition is by the sequence/phase and duty factor that the feet are in contact with the ground. Alternatively, center of mass (CoM) dynamics can be used to define gait instead. With these different ways, what appear as walking (e.g., cockroach using an alternating tripod footfall pattern with a >0.5 duty factor) using the first definition may in fact be classified as SLIP-like running when CoM dynamics are evaluated.The authors have carefully compared their definition of gait with that in a few closely-related recent studies. The authors' gait definition is mainly based on kinematics (coordination to be exact) of the legs themselves, whereas the other studies used stance and swing phase patterns (which are footfall patterns on the ground). These are different ways of defining gait from kinematics. While I agree with the technical aspects of the comparisons, as well as some flaws that the authors have identified in the other studies, I am not sure if leg coordination alone is THE best/right way of defining gait.As an example, we can again consider the cockroach case above. Using the author's definition, cockroaches also only demonstrates a single tripod "gait" in terms of leg coordination when speed increase from very small to very large. Does this mean that all the work defining cockroach walking at low speeds and running at high speeds using CoM dynamics are wrong in having such definitions?In my opinion, the authors have shown convincing evidence that leg coordination (which they refer to as "gait") does not change significantly as speed increases. The resulting kinematics of stance and swing phases do vary with speed, and other researchers (or the authors hypothetically) may choose to use those instead to define gait. Frankly, I do not think this is the key advancement of the study and am not sure it is worth this much effort to try to convince people of different opinions (although I certainly understand that the authors may be choosing to emphasize this in order to address the reviewers' comments).

We completely agree with the reviewer. As such we have stated in the first paragraph itself that we are using gait interchangeably with inter-leg coordination. We have reformatted the manuscript to remove much of the description of gait.

I think the authors should highlight the contribution they are making with the ARSLIP model. Again, as I said above, this can potentially be a general model and a major advancement of the field of terrestrial animal locomotion. I see this model (with further validation and generalization beyond this study) as the equivalent of seminal work on bipedal compliant leg template model, which unifies SLIP running and inverted pendulum walking for bipedal locomotion (Geyer, Seyfarth and Blickhan, 2006). The authors cited this work, but do not seem to appreciate its importance in the area of bipedal locomotion, or how their own work has the potential of being the same. The authors' model provides initial insight why hexapedal animals can use the same set of legs to achieve dynamic locomotion over a wide range of speeds.The authors do provide evidence that the velocity profile of CoM dynamics are opposite to SLIP-like, which is another thing that motivates the ARSLIP model. I think this should be also emphasized more to set up the model.I think the authors are missing an opportunity not to discuss this more, and their paper will improve substantially by building around this central idea, which the kinematic data and analysis provide compelling evidence for. I think this will significantly increase the impact of the paper beyond the field of animal locomotion. It is these kinds of simple, general biomechanical/dynamic models that have provided the foundation for simple yet robust robots such as RHex, Atlas, BigDog, etc. Personally, I think that this is something that many people will remember the study for, not so much the technical debate on which way of defining "gait" from pure kinematics is better, which are concerns more for the specialists.Therefore, I strongly encourage the authors to cut down on the debate of "gait" and highlight the modeling contributions and elaborate (at least speculate) what should be done in future to test and validate it as a general model.

We have addressed this point by embracing the idea that ARLSIP could be a general model for multi-legged locomotion (this includes the double stance phase of bipedal locomotion to quadrupeds to hexapods and polypeds) through a more detailed discussion in both the Introduction and Discussion.